# SINGLE SMPC INVOCATION DPHELMET: DIFFERENTIALLY PRIVATE DISTRIBUTED LEARNING ON A LARGE SCALE

## ABSTRACT

We introduce a distributing differentially private machine learning training protocol that locally trains support vector machines (SVMs) and computes their averages using a single invocation of a secure summation protocol. With state-of-the-art secure summation protocols and using a strong foundation model such as SimCLR, this approach scales to a large number of users and is applicable to non-trivial tasks, such as CIFAR-10. Our experimental results illustrate that for 1,000 users with 50 data points each, our scheme outperforms state-of-the-art scalable distributed learning methods (differentially private federated learning, short DP-FL) while requiring around 500 times fewer communication costs: For CIFAR-10, we achieve a classification accuracy of $79.7\,\%$ for an $\varepsilon = 0.59$ while DP-FL achieves $57.6\,\%$. More generally, we prove learnability properties for the average of such locally trained models: convergence and uniform stability. By only requiring strongly convex, smooth, and Lipschitz-continuous objective functions, locally trained via stochastic gradient descent (SGD), we achieve a strong utility-privacy trade-off.

## 1 INTRODUCTION

Scalable distributed privacy-preserving machine learning methods have a plethora of applications, ranging from medical institutions that want to learn from distributed patient data, over edge AI health applications, to decentralized recommendation systems. Preserving each person's privacy during distributed learning raises two challenges: (1) during the distributed learning process the inputs of all parties have to be protected and (2) the resulting model itself should not leak information about the contribution of any person to the training data. To tackle (1), secure multi-party computation protocols (SMPC) can protect data during distributed computation. To tackle (2), differentially private (DP) mechanisms provide guarantees for using or releasing the model in a privacy-preserving manner.

The literature contains a rich body of work on this kind of privacy-preserving distributed machine learning (PPDML) which is frequently evaluated with respect to scalability with the number of users who participate in the distributed learning, expressivity of the learning method with the goal of encompassing complex learning tasks, and a good utility-privacy trade-off without a significant loss in accuracy for protecting each person's data, optimally the same utility-privacy trade-off as the centralized training scheme while only adding little communication overhead.

Jayaraman et al. (2018) introduced a theoretic result where the model optimum is noised (output perturbation). Here, each of the $n$ users locally trains a convex empirical risk minimization (ERM) model on $m$ data points and contributes the parameters of this model, carefully noised to a single invoked SMPC step, resulting in an averaged differentially private model. This approach achieves DP (Chaudhuri et al., 2011), requires as little noise as the centralized setting ($\mathcal{O}(1/nm)$), and incurs little communication overhead, with one SMPC invocation. However, they use untight utility bounds Pathak et al. (2010) that scale with the number of local data points ($\mathcal{O}(1/m)$) and not with the combined number of data points across all users ($\mathcal{O}(1/nm)$).

Jayaraman et al. (2018) prove strong utility bounds with another scheme, the gradient perturbation: each user contributes the gradients of each local training iteration carefully noised to a single invoked SMPC step which results in an averaged differentially private gradient step. This construction adds as little noise as centralized training ($\mathcal{O}(1/nm)$) and achieves strong utility bounds which scale with

the number of data points across all users ($\mathcal{O}(1/nm)$). However, it has considerable communication overhead since it requires one SMPC invocation per training iteration.

Federated learning (McMahan et al., 2017) with a DP-SGD approximation (Abadi et al., 2016) (DP-FL) constitutes another line of research with moderate utility bounds and moderate communication overhead. In DP-FL, an untrusted aggregator combines the gradient updates from each user, while each user satisfies DP. DP-FL does not require SMPC for similar security guarantees but needs $\mathcal{O}(\#training\_steps)$ communication rounds. The utility bounds are comparatively high since the noise scales with $\mathcal{O}(m\sqrt{n})$. Appx. C discusses related work in more detail.

Concerning expressivity, Abadi et al. (2016); Tramèr & Boneh (2021); De et al. (2022) have shown that pre-trained models can improve the performance of a differentially private machine learning method (DP-SGD) for non-trivial tasks (e.g., CIFAR-10). While such models require sufficient public data, they exist and provide simplifying representations for various domains: SimCLR for pictures, Facenet for portrait pictures, UNet for medical segmentation imagery, or GPT-3 for natural language.

Yet, this prior work does not excel at all three metrics simultaneously: scalability, expressivity, and utility-privacy trade-off. This places an inherent disadvantage when comparing current distributed training processes to a centralized training process.

**Contributions.** Our Secure Distributed DP-Helmet work extends on prior work (Jayaraman et al., 2018) such that it is scalable, expressive, and has a good utility-privacy trade-off. Table 1 compares our approach with Jayaraman et al. (2018)'s approaches and DP-FL. In summary, we make two tangible contributions:

1. For SGD-based strongly convex ERM, we prove a tighter utility bound which essentially states that we only need the average of locally trained models, e.g. support vector machines (SVMs) or logistic regression (LR), to converge to the optimal centrally trained model with rate $\mathcal{O}(1/M)$ for $M$ iterations (cf. Thm. 21). We also show train-test generalization by proving uniform stability which states that averaging our models linearly improves the stability bound (cf. Thm. 19).

2. In Cor. 10 we show how with enough data, guarantees as in local DP can be achieved, even without assumptions on the training algorithm beyond a norm-bounded parameter space: we protect the entire input of a user while achieving strong utility bounds ($> 80\%$ test accuracy for CIFAR-10).

## 2 OVERVIEW

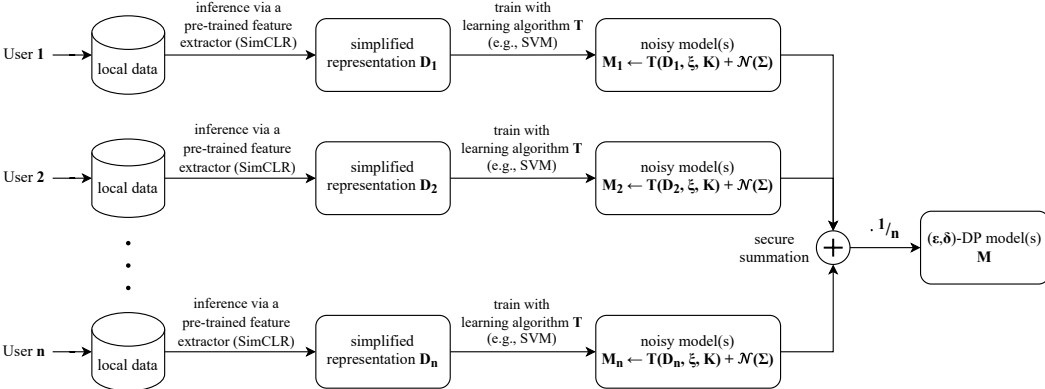

Figure 1: Schematic overview of Secure Distributed DP-Helmet. Each user locally extracts a simplified data representation via a pre-training feature extractor (SimCLR), then trains a model, e.g. an SVM, via a learning algorithm $T$, and finally contributes a model which is carefully noised with a spherical $\Sigma$-parameterized Gaussian to a single invoked secure summation step which results in an averaged and $(\varepsilon, \delta)$-DP model. $\xi$ denotes some hyperparameters and $K$ a set of classes.

**Systems overview.** Secure Distributed DP-Helmet achieves scalable, distributed privacy-preserving training on sensitive data with a strong classification performance. A schematic overview

of our work is illustrated in Fig. 1. Here each user – the protocol party – holds and protects a small dataset while all users jointly learn a model without leaking information about the local dataset. More precisely, the jointly computed model protects information about individual persons in two scenarios: first, each user is a local aggregator (e.g., a hospital) and each person contributes one data point (differential privacy, see Fig. 3); second, each user is a person and contributes a small dataset (local DP, see Fig. 2 for $\Upsilon = 50$).

Essentially, we base our method on the following simple yet effective scheme introduced by Jayaraman et al. (2018): Each user locally trains a model, e.g. an SVM, via a learning algorithm $T$, and contributes the parameters of this model carefully noised to a single invoked secure summation or SMPC step which results in an averaged and $(\varepsilon, \delta)$-DP model. This construction allows using only $\mathcal{O}(1/nm)$ much noise for $n$ users with $m$ data points each – the same as in a centralized setting.

Thus we seek and subsequently achieve three criteria: (1) scalability with the number of users which is measured by the number of communication rounds and the number of secure summation invocations if required with the above privacy definition. (2) high expressivity which is measured by the classification accuracy as well as the utility degradation when compared to a centralized scheme where all data is stored at the aggregator. (3) good utility-privacy trade-off which is measured by how much more noise we need to add due to the distributed training scheme.

**Related Work.** In Table 1 we detail our utility-bound improvement in comparison to prior work. Observe that our work matches the utility bound and privacy bound of centralized training while having constant secure summation overhead: We only require one invocation of secure summation which is even realistic for Smartphone-based applications since we do not have to deal with issues of multiple consecutive communication rounds like dropouts or unstable connectivity. When using Bell et al. (2020)'s construction for secure summation, the number of communication rounds is fixed to 4 rounds and the size of each communication round is increased by only $\log(n\_users)$, which is diminishingly small when compared to the constant overhead amounting to the model size. In comparison to DP-FL, we have a 500-fold decrease in communication cost: DP-FL has 1,920 rounds of size $\ell$, where $\ell$ is the model size (roughly 60,000 floats for CIFAR-10) while we have 4 rounds of size $\log(n\_users) + \ell$ for roughly the same model size.

**Evaluation.** Our evaluation on CIFAR-10 with SimCLR-based pre-training shows that for 1,000 users with 50 data points per user, our scheme achieves with SVMs a classification accuracy of 79.7 % for an $\varepsilon = 0.59$. Extrapolated to hundreds of thousands and millions of protocol parties (see Fig. 2 for a thorough evaluation), we can protect the entire local dataset (say, 50-group DP) of a protocol party and achieve high accuracy: For local datasets of size 50, Secure Distributed DP-Helmet achieves $\geq 84$ % accuracy and $(\varepsilon, \delta) = (0.01, 10^{-10})$ for $200,000$ users and $\geq 87$ % accuracy and $(\varepsilon, \delta) = (2 \cdot 10^{-4}, 10^{-12})$ for 20,000,000 users while guaranteeing local DP-like properties (or 50-group DP). In our experiments, Secure Distributed DP-Helmet significantly improves upon DP-FL for more than $400 \leq n$ users (for CIFAR-10), as it avoids the factor $\sqrt{n}$ noise overhead.

Table 1: Comparison to related work for $n$ users with $m$ data points each: utility guarantee to the population optimum, DP noise scale, and number of SMPC invocations. In DP-FL an untrusted aggregator combines the updates from each user, while each user update satisfies DP (by adding noise and norm-clipping each gradient). It does need a communication round per training iteration $M$.

| Algorithm | Utility | Noise scale | SMPC Invocations |
|---|---|---|---|
| DP federated learning (DP-FL) | $\mathcal{O}(1/nm)$ | $\mathcal{O}(1/m\sqrt{n})$ | $-$ ($\mathcal{O}(M)$ rounds) |
| Jayaraman et al. (2018), gradient perturbation | $\mathcal{O}(1/nm)$ | $\mathcal{O}(1/nm)$ | $\mathcal{O}(M)$ |
| Jayaraman et al. (2018), output perturbation | $\mathcal{O}(1/m)$ | $\mathcal{O}(1/nm)$ | 1 |
| *Secure Distributed DP-Helmet (ours)* | $\mathcal{O}(1/nm)$ | $\mathcal{O}(1/nm)$ | 1 |
| Centralized training | $\mathcal{O}(1/nm)$ | $\mathcal{O}(1/nm)$ | 0 |

## 2.1 KEY IDEAS

**Trustworthy distributed noise generation.** One core requirement of SMPC-based distributed learning is honestly generated and unleakable noise as otherwise, our privacy guarantees would not hold anymore. There is a rich body of work on distributed noise generation (Moran et al., 2009;

Dwork et al., 2006a; Kairouz et al., 2015b; 2021; Goryczka & Xiong, 2015). So far, however, no distributed noise generation protocol scales to millions of users. To jointly create noise of a given magnitude, we can alternatively use a simple, yet effective technique: utilizing the large number of users in our system, we can reasonably assume that at least a fraction of them (say $t = 50\%$) are not colluding to violate privacy by sharing the noise they generate with each other. As long as we combine the noise of each user in an oblivious fashion, every user can create noise separately and independently and we are still guaranteed noise of a magnitude depending on $t$.

**Strong utility-privacy tradeoff via tight composition and convexity.** Secure Distributed DP-Helmet utilizes differentially private SGD-based SVMs for which strong utility-privacy tradeoffs have been shown (Wu et al., 2017). We use SVMs for multi-class classification via the one-vs-rest (OVR) scheme. Each class is waged against the combination of all other classes during training. We rely on output perturbation to estimate a sensitivity bound on the resulting models and add calibrated noise to the model. Meiser & Mohammadi (2018); Sommer et al. (2019); Balle et al. (2020a) show tight composition bounds for such sensitivity-bounded additive mechanism.

**Expressivity of our approach.** A known limitation of convex SVMs is their limited expressivity. As a remedy, we utilize transfer learning and operate in two phases: a pre-training phase in which a powerful representation model is trained on a public dataset, and a training phase in which we train a set of SVMs on a sensitive dataset. The datasets can have different, disjoint distributions; it suffices that the two datasets are comparable in structure. In our evaluation, we use the SimCLR representation model (Chen et al., 2020b) trained on ImageNet and then fine-tune it on CIFAR-10.

**Threat model & security goals.** We separate the security assumptions of our protocol and those of the underlying secure summation. For our work, we assume security against malicious, global attackers that do not follow our protocol as long as we have a ratio of at least $t$ honest users (say $t = 50\,\%$). In particular, we consider dishonest noise generation. The attacker in both variants tries to extract sensitive information about other parties from the interaction and the result. As in other strong security definitions, the attacker has strong background knowledge and knows everything about and can influence each user's dataset, except for one data point of one user. Our privacy goals are $(\varepsilon, \delta)$-differential privacy (protecting single samples) and $(\varepsilon, \delta)$-$\Upsilon$-*group differential privacy* (protecting all samples of a user at once).

## 2.2 WHAT DOES THAT MEAN FOR PRACTICAL APPLICATIONS?

From a bird's eye view, our experiments show that the mean of several SVMs significantly improves the classification accuracy, even if the individual SVMs have a poor performance on their own. We pushed this approach to its limit and trained each SVMs on only 50 data points. Our experiments show that the SVM obtained by computing the mean of 1,000 such SVMs has very high accuracy.

**Local DP.** For applications with around 200,000 or more users, we can protect the entire local dataset of a user, i.e., the entire locally trained SVM. This result can be generalized: We can protect each local SVM independently of how many data points were used to train it. With this generalized view, our scheme does not only protect data points but users, which makes our DP guarantees akin to a group-DP setting (comparable to local DP). Applications can leverage this method and let users train SVMs on their own devices instead of requiring local aggregators for sets of users.

**For which other learning algorithms is this framework applicable?** Our utility-privacy results apply, beyond SGD-based SVMs, to other learning methods; those methods improve significantly upon averaging locally trained models (cf. Cor. 10 and Appxs. M and N). We show that the following five requirements suffice for showing both differential privacy and learnability properties: (1) bounded output sensitivity, (2) strongly convex training objective, (3) smooth training objective, (4) Lipschitz continuous training objective, and (5) SGD-based update routine. Notably, we formally only require that the norm of each model is bounded; we do not make any assumptions about the training procedure of each base learner, which learns a single model in the ensemble. In particular, base learners do not need to satisfy differential privacy. Concerning learnability properties, we show that a generalizability property and a convergence property improve the average of locally trained models when compared to the locally trained models, for a certain class of learning methods. Whenever SGD is used with a strongly convex, smooth, and Lipschitz ERM objective (empirical risk minimization), averaging local models improves on uniform stability, which is a form of generalizability property, and convergence, which measures the loss-distance to the optimal model for a given dataset.

---

**Algorithm 1:** SGD_SVM$(D, \xi, K)$ with hyperparameters $\xi := (h, c, \Lambda, R, M)$

---

**Data:** dataset $D := \{(x_i, y_i)\}_{i=1}^N$ where $x_i$ is structured as $[1, x_{i,1}, \ldots, x_{i,p}]$;   set of classes $K$;
  Huber loss smoothness parameter $h \in \mathbb{R}_+$;    input clipping bound: $c \in \mathbb{R}_+$;   #iterations $M$;
  regularization parameter: $\Lambda \in \mathbb{R}_+$;           model clipping bound: $R \in \mathbb{R}_+$;

**Result:** models (1d intercepts with $p$-dimensional hyperplanes): $\left\{ f_M^{(k)} \right\}_{k \in K} \in \mathbb{R}^{(p+1) \times |K|}$

---

1 $clipped(x) := c \cdot x / \max(c, \|x\|)$;
2 $\mathcal{J}(f, D, k) := \frac{\Lambda}{2} f^T f + \frac{1}{N} \sum_{(x,y) \in D} \ell_{huber}\left(f^T clipped(x) \cdot y \cdot (1[y = k] - 1[y \neq k])\right)$;
3 **for** $k$ **in** $K$:
4   **for** $m$ **in** $1, \ldots, M$:
5     $f_m^{(k)} \leftarrow SGD(\mathcal{J}(f_m, D, k), \alpha_m)$, with learning rate $\alpha_m := \min(\frac{1}{\beta}, \frac{1}{\Lambda m})$ and
      $\beta = 1/2h + \Lambda$;
6     $f_m^{(k)} := R \cdot f_m^{(k)} / \|f_m^{(k)}\|$;                          `// projected SGD`

---

## 3 PRELIMINARIES

### 3.1 DIFFERENTIAL PRIVACY AND DP_SGD_SVM

Preliminaries of the Secure Summation protocol (Bell et al., 2020) as well as pre-training as a tactic to boost DP performance are available in Appx. B.2 and Appx. B.3 respectively.

As a privacy notion, we consider differential privacy (DP) (Dwork et al., 2006b). Intuitively, differential privacy quantifies the protection of any individual's data within a dataset against an arbitrarily strong attacker observing the output of a computation on said dataset. Strong protection is achieved by bounding the influence of each individual's data on the resulting SVMs. For the (standard) definition of differential privacy we utilize in our proofs, we refer to Appx. B.1.

We consider Support Vector Machines (SVMs), which can be made strongly convex, thus display a unique local minimum, and have a lower bound on the growth of the optimization function. Having a unique local minimum makes those methods ideal for computing tight differential privacy bounds and thus highly relevant machine learning predictors for our work. In fact, this differentially private SVM definition (DP_SGD_SVM) can be derived directly from the work of Wu et al. (2017) on empirical risk minimization using SGD-based optimization. They rely on a smoothed version of the hinge-loss: the Huber loss $\ell_{huber}$ (cf. Appx. B.4 for details). We additionally apply norm-clipping to all inputs. We use the one-vs-rest (OVR) method to achieve a multiclass classifier. Alg. 1 provides pseudocode for the sensitivity-bounded algorithm (before adding noise).

In contrast to Wu et al. (2017), which assumes for each data point $\|x\| \leq 1$, we use a generalization that holds for larger norm bounds $c > 1$: we assume $\|x\| \leq c$, where $c$ is a hyperparameter of the learning algorithm SGD_SVM. As a result, the optimization function $\mathcal{J}$ is $c + R\Lambda$ Lipschitz (instead of $1 + R\Lambda$ Lipschitz as in Wu et al. (2017)) and $((c^2/2h + \Lambda)^2 + p\Lambda^2)^{1/2}$ smooth (instead of $1/2h + \Lambda$ smooth). Wu et al. (2017) showed a *sensitivity bound* for SGD_SVM from which we can conclude DP guarantees. The sensitivity proof follows from Wu et al. (2017, Lemma 8) with the Lipschitz constant $L = c + R\Lambda$, a smoothness $\beta = ((c^2/2h + \Lambda)^2 + p\Lambda^2)^{1/2}$ and a $\Lambda$-strong convexity.

Similarly, our work applies to $L_2$-regularized logistic regression where we adapt Alg. 1 with the optimization function $\mathcal{J}'(f, D) := \frac{\Lambda}{2} f^T f + \frac{1}{N} \sum_{(x,y) \in D} \ln(1 + \exp(-f^T clipped(x) \cdot y))$ which is $\Lambda$-strongly convex, $L = c + R\Lambda$ Lipschitz, and $\beta = ((c^2/4 + \Lambda)^2 + p\Lambda^2)^{1/2}$ smooth. We would also need to adapt the learning rate to accommodate the change in the smoothness parameter but continue to have the same sensitivity as for the classification case.

**Definition 1** (Sensitivity). *Let $f$ be a function that maps datasets to the $p$-dimensional vector space $\mathbb{R}^p$. The* sensitivity *of $f$ is defined as $\max_{D \sim_1 D'} \|f(D) - f(D')\|$, where $D \sim_1 D'$ denotes that the datasets $D$ and $D'$ differ in at most one element. We say that $f$ is an $s$-sensitivity-bounded function.*

The following lemma directly follows from Wu et al. (2017, Lemma 8).

**Lemma 2.** *With the input clipping bound $c$, the model clipping bound $R$, the strong convexity factor $\Lambda$, and the number of data points $N$, the learning algorithm SGD_SVM of Alg. 1 has a sensitivity bound of $s = \frac{2(c+R\Lambda)}{N\Lambda}$ for each of the $|K|$ output models.*

For sensitivity-bounded functions, there is a generic additive mechanism that adds Gaussian noise to the results of the function and achieves differential privacy, if the noise is calibrated to the sensitivity.

**Lemma 3** (Gaussian mechanism is DP (Theorem A.1 & Theorem B.1 in Dwork & Roth (2014)))**.** *Let $q_k$ be functions with sensitivity $s$ on the set of datasets $\mathcal{D}$. For $\varepsilon \in (0,1)$, $c^2 > 2\ln 1.25/(\delta/|K|)$, the Gaussian Mechanism $D \mapsto \{q_k(D)\}_{k \in K} + \mathcal{N}(0, (\sigma \cdot I_{(p+1)\times|K|})^2)$ with $\sigma \geq \frac{c \cdot s \cdot |K|}{\varepsilon}$ is $(\varepsilon, \delta)$-DP, where $I_d$ is the $d$-dimensional identity matrix.*

The mechanism that first learns $|K|$ SVM models via SGD_SVM of Alg. 1 and then adds multivariate Gaussian Noise $\mathcal{N}(0, (\sigma \cdot I_{(p+1)\times|K|})^2)$ is DP. Note that there are tighter composition results (Meiser & Mohammadi, 2018; Sommer et al., 2019; Balle et al., 2020a) where $\varepsilon \in \mathcal{O}(\sqrt{|K|})$ which we do not formalize for brevity reasons but follow in our experiments.

**Corollary 4** (Gaussian mechanism on SGD_SVM is DP)**.** *With the $s$-sensitivity-bounded learning algorithm SGD_SVM (cf. Lem. 2), the dimension of each data point $p$, the set of classes $K$, and $\varepsilon \in (0,1)$, DP_SGD_SVM$(D, \xi, K, \sigma) \coloneqq$ SGD_SVM$(D, \xi, K) + \mathcal{N}(0, (\sigma \cdot s \cdot I_{(p+1)\times|K|})^2)$ is $(\varepsilon, \delta)$-DP, where $\varepsilon \geq \sqrt{2\ln 1.25/(\delta/|K|)} \cdot |K| \cdot 1/\sigma$ and $I_d$ is the $d$-dimensional identity matrix.*

**Notation.** A *learning algorithm* is a function from datasets to learned models. Subsequently, we consider the notion of a configuration in many theorems.

**Definition 5** (Configuration $\zeta$)**.** *A configuration $\zeta(\mathcal{U}, t, T, s, \xi, \mho, i, N, K, \sigma)$ consists of a set of users $\mathcal{U}$ of which $t \cdot \mathcal{U}$ are honest, an $s$-sensitivity-bounded learning algorithm $T$ on inputs $(D, \xi, K)$, hyperparameters $\xi$, a local datasets $D^{(i)}$ of user $U^{(i)} \in \mathcal{U}$ with $N = \min_{i \in \{1,...,|\mathcal{U}|\}} |D^{(i)}|$ and $\mho = \bigcup_i^{|\mathcal{U}|} D^{(i)}$, a set of classes $K$, and a noise multiplier $\sigma$. $avg(T)$ is the aggregation of $|\mathcal{U}|$ local models of algorithm $T$: $avg(T(\mho)) = \frac{1}{|\mathcal{U}|} \sum_{i=1}^{|\mathcal{U}|} T(D^{(i)}, \xi, K)$. If unique, we simply write $\zeta$.*

## 4 SECURE DISTRIBUTED DP-HELMET

This section presents Secure Distributed DP-Helmet in detail (cf. Alg. 2) and its utility-privacy properties. Here, each user separately trains a sensitivity-bounded learning algorithm, e.g. DP_SGD_SVMs, before their parameters are combined with the parameters trained by other users via a single round of secure summation. The single round of secure multiparty computation allows us to have the full benefit of securely aggregating data: we can show centralized-DP guarantees within a threat model akin to that of federated learning with differential privacy.

**Active attacks and untrustworthy noise.** Our threat model allows each user to place very little trust in other users. However, we focus on passive adversaries. Active attacks that, e.g., aim to poison the resulting model, are left for future work. Note that even passive adversaries can collude and exchange information about the randomness they used in their local computation. As we combine the noise added by different users, we need to take into account that not all of that noise is necessarily secret to the adversary. To compensate for untrustworthy users, we double the noise added by each user; as long as half of all users are honest, our guarantees thus are valid.

Next, we derive a tight output sensitivity bound. A naïve approach would be to release each individual predictor, determine the noise scale proportionally to $\tilde{\sigma} \coloneqq \sigma$ (cf. Cor. 4), showing $(\varepsilon, \delta)$-DP for every user. We can save a factor of $|\mathcal{U}|^{1/2}$ by leveraging that $|\mathcal{U}|$ is known to the adversary and we have at least $t = 50\%$, yielding $\tilde{\sigma} \coloneqq \sigma \cdot 1/\sqrt{t \cdot |\mathcal{U}|}$.

**Corollary 6.** *Given a configuration $\zeta$, Secure Distributed DP-Helmet$(\zeta)$ (cf. Alg. 2) without adding noise, i.e. $avg(T(\mho))$, has a sensitivity of $s \cdot 1/|\mathcal{U}|$ for each class $k \in K$.*

The proof is placed in Appx. G. Having bounded the sensitivity of the aggregate to $s \cdot 1/|\mathcal{U}|$, we show that locally adding noise per user proportional to $\sigma \cdot s \cdot 1/\sqrt{|\mathcal{U}|}$ and taking the mean is equivalent to only centrally adding noise proportional to $\sigma \cdot s \cdot 1/|\mathcal{U}|$ (as if the central aggregator was honest).

**Lemma 7.** *Given a configuration $\zeta$ and any noise scale $\tilde{\sigma}$, then $\frac{1}{|\mathcal{U}|} \sum_{i=1}^{|\mathcal{U}|} \mathcal{N}(0, (\tilde{\sigma} \cdot 1/\sqrt{|\mathcal{U}|})^2) = \mathcal{N}(0, (\tilde{\sigma} \cdot 1/|\mathcal{U}|)^2)$.*

**Algorithm 2:** Secure Distributed DP-Helmet. For $T = $ SGD_SVM (cf. Alg. 1) we have $s = \frac{2(c+R\Lambda)}{N\Lambda}$ with hyperparameters $\xi := (h, c, \Lambda, R, M)$. $\pi_{SecAgg}$ is described in Bell et al. (2020, Algorithm 2) and can be extended to floating points using fixed-point arithmetic.

---

**def** Client Secure Distributed DP-Helmet (*D, |U|, K, T, t, ξ, σ*)**:**

  **Data:** local dataset $D$ with $N = |D|$; #users $|\mathcal{U}|$; set of classes K; training algorithm $T$; ratio $t$ of honest users; hyperparameters $\xi$; noise multiplier $\sigma$

  **Result:** DP-models (intercepts with $p$-dimensional hyperplanes): $M_{priv} := \left\{ f_{priv}^{(k)} \right\}_{k \in K}$

1 | $M \leftarrow T(D, \xi);$            // T is $s$-sensitivity-bounded
2 | $M_{priv} \leftarrow M + \mathcal{N}(0, (\tilde{\sigma} \cdot s \cdot I_{p+1 \times |K|})^2)$ **with** $\tilde{\sigma} := \sigma \cdot 1/\sqrt{t \cdot |\mathcal{U}|};$
3 | Run the client code of the secure summation protocol $\pi_{SecAgg}$ on input $M_{priv}/|\mathcal{U}|;$

**def** Server Secure Distributed DP-Helmet (*U*)**:**

  **Data:** users $\mathcal{U}$
  **Result:** empty string

4 | Run the server protocol of $\pi_{SecAgg};$

---

The proof is placed in Appx. H. We can now prove differential privacy for Secure Distributed DP-Helmet of Alg. 2 where we have noise scale $\tilde{\sigma} := \sigma \cdot 1/\sqrt{t \cdot |\mathcal{U}|}$ and thus $\varepsilon \in \mathcal{O}(s/\sqrt{t \cdot |\mathcal{U}|})$.

**Theorem 8** (Main Theorem, simplified). *Given a configuration $\zeta$, Secure Distributed DP-Helmet($\zeta$) (cf. Alg. 2) satisfies computational $(\varepsilon, \delta + \nu)$-DP with $\varepsilon \geq \sqrt{2 \ln 1.25/(\delta/|K|)} \cdot |K| \cdot 1/\sigma$ and a function $\nu$ negligible in the security parameter of the secure aggregation.*

The full statement and proof are in Appx. I. Simplified, the proof follows by the application of the sensitivity (cf. Cor. 6) to the Gauss Mechanism (cf. Lem. 3) where the noise is applied per user (cf. Lem. 7). If each user contributes 50 data points and we have 1000 users, $N \cdot |\mathcal{U}| = 50,000$.

Next, we show that we can protect the entire dataset of a single user (e.g., for distributed training via smartphones). The sensitivity-based bound on the Gaussian mechanism (see Appx. K) directly implies that we can achieve strong $\Upsilon$-group privacy results, which is equivalent to local DP.

**Corollary 9** (Group-private variant). *Given a configuration $\zeta$, Secure Distributed DP-Helmet($\zeta$) (cf. Alg. 2) satisfies computational $(\varepsilon, \delta + \nu)$, $\Upsilon$-group DP with $\varepsilon \geq \Upsilon \cdot \sqrt{2 \ln 1.25/(\delta/|K|)} \cdot |K| \cdot 1/\sigma$ for $\nu$ as above: for any pair of datasets $\mho, \mho'$ that differ at most $\Upsilon$ many data points,*

*Secure Distributed DP-Helmet$(\zeta(\dots, \mho, \dots)) \approx_{\varepsilon, \delta}$ Secure Distributed DP-Helmet$(\zeta(\dots, \mho', \dots))$*

**Cor. 9 and DP-FL.** While the averaging will slightly offset this massive amount of noise, such a result does not hold for DP-FL because in the local training the sensitivity does not decrease. Hence, in contrast to Secure Distributed DP-Helmet, the standard deviation of the noise that is locally added will continuously increase, no matter how many users join the distributed training.

Cor. 9 generalizes to a more comprehensive Cor. 10 that is data oblivious. If we can show that the training algorithm of every user has the same bounded sensitivity, i.e., that the norm of each model is bounded, then Secure Distributed DP-Helmet can apply to the granularity of users instead of that of data points. We explicitly don't need to make any further assumptions about the training procedure of each base learner; it is sufficient that the local models are combined via noisy arithmetic mean. This method renders a tighter sensitivity bound than SGD_SVM for certain settings of $\Upsilon$ or data points per user $N$. Moreover, it enables the use of other SVM optimizers or Logistic Regression.

In particular, the training procedure of each base learner does not need to satisfy differential privacy.

**Corollary 10.** *Given a learning algorithm $T$, we say that $T$ is $R$-norm bounded if for any input dataset $D$ with $N = |D|$, any hyperparameter $\xi$, and all classes $k \in K$, $\|T(D, \xi, k)\| \leq R$. Any $R$-norm bounded learning algorithm $T$ has a sensitivity $s = 2R$. In particular, $T + \mathcal{N}(0, (\sigma \cdot s \cdot I_d)^2)$ satisfies $(\varepsilon, \delta)$, $\Upsilon$-group differential privacy with $\Upsilon = N$ and $\varepsilon \geq \sqrt{2 \ln 1.25/(\delta/|K|)} \cdot |K| \cdot 1/\sigma$, where $\mathcal{N}(0, (\sigma \cdot s \cdot I_d)^2)$ is spherical multivariate Gaussian noise and $\sigma$ a noise multiplier.*

The proof is in Appx. J. Here the number of local data points $N$ can vary among the users.

STABILITY & CONVERGENCE. The core of our approach is to locally train models and compute the average without further synchronizing or fine-tuning of the models: $\mathrm{avg}(T)$. For $T = \mathrm{SGD\_SVM}$, we prove that the learnability properties (Shalev-Shwartz et al., 2010) uniform stability and convergence are comparable to a centrally trained SGD_SVM.

**Uniform stability.** We show in Appx. M that the training generalizes well by proving uniform stability in the sense of Bousquet & Elisseeff (2002) for $T = \mathrm{SGD\_SVM}$: $|\mathbb{E}[\mathcal{J}(\mathrm{avg}(T(\mho)), \mho, \_) - \mathbb{E}_{z \in \mathcal{Z}}[\mathcal{J}(\mathrm{avg}(T(\mho)), z, \_)]]| \in \mathcal{O}(|\mho|^{-1})$ where $\mathcal{J}$ is the objective function (cf. Alg. 1) and $\mathcal{Z}$ an unknown data distribution where $\mho \in \mathcal{Z}$. In particular, we show that averaging the locally trained SGD_SVMs linearly improves the stability bound.

**Convergence.** In line with Zhang et al. (2013) on averaged ERM models, we show in Appx. N that avg(SGD_SVM) gracefully converges to the best model for the combined local datasets $\mho$: $\mathbb{E}[\mathcal{J}(\mathrm{avg}(\mathrm{SGD\_SVM}(\mho)), \mho, \_) - \inf_f \mathcal{J}(f, \mho, \_)] \in \mathcal{O}(1/M)$ for $M$ many training iterations.

## 5 EXPERIMENTAL RESULTS

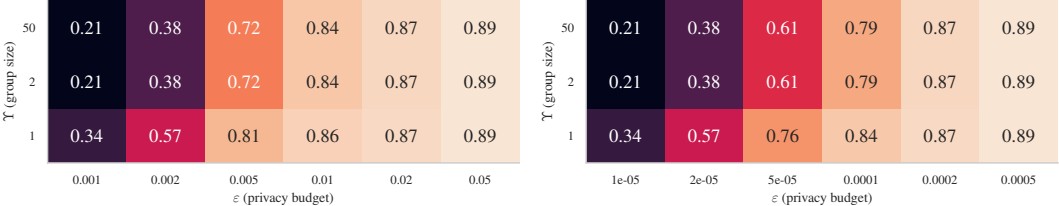

Figure 2: $(\varepsilon, \Upsilon)$-Heatmap for classification accuracy of Secure Distributed DP-Helmet on CIFAR-10 dataset (left: $\delta = 10^{-10}$; right: $\delta = 10^{-12}$) with 200,000 (left) and 20,000,000 (right) users. We train 1,000 models on 50 data points each; to emulate having more users we rescale the $\varepsilon$-values ($\varepsilon' := 1000 \cdot \varepsilon \cdot \Upsilon/n_{users}$) and report the respective (interpolated) accuracy values. We extrapolate the privacy guarantees, due to the limited dataset size. Our accuracy values are pessimistic as we keep the accuracy numbers that we got from averaging 1,000 models. Actually taking the mean over 200,000 or even 20,000,000 users should provide better results. In our evaluation, $\Upsilon = 50$ group privacy is comparable to local DP. A lower value of $\Upsilon < 50$ places trust in users as local aggregators. For $\Upsilon \geq 2$, we can use a tighter group-privacy bound (cf. Cor. 10); hence, the accuracy values are the same as for $\Upsilon = 50 = N$, where the entire local data set of a user is protected.

We analyze four experimental questions: (RQ1) First, how does Secure Distributed DP-Helmet as well as the strongest alternative, DP-SGD-based federated learning, perform in terms of privacy-utility trade-off? Moreover, how does the performance change if we allow more users (cf. Fig. 3, right, and Fig. 4)? (RQ2) Second, what is the utility loss of applying both methods in a distributed fashion instead of centrally (cf. Fig. 3, left)? (RQ3) Third, how does Secure Distributed DP-Helmet perform when we have truly many users ($\geq$ 200,000 users) and when we are in a local-DP setting (cf. Fig. 2)? (RQ4) Fourth, how do learning algorithms different than DP_SGD_SVM perform (cf. Appx. F)?

**Pretraining.** We used a SimCLR pre-trained model[1] on ImageNet ILSVRC-2012 (Russakovsky et al., 2015) for all experiments (cf. Fig. 6 in the appendix for an embedding view). It is built with a ResNet152 with selective kernels (Li et al., 2019) architecture including a width multiplier of 3 and it has been trained in the fine-tuned variation of SimCLR where $100\,\%$ of ImageNet's label information has been integrated during training. Overall, it totals $795\,M$ parameters and achieves $83.1\,\%$ classification accuracy (1000 classes) when applied to a linear prediction head. In comparison, a supervised-only model of the same size would only achieve $80.5\,\%$ classification accuracy.

**Sensitive Dataset.** CIFAR-10 (Krizhevsky, 2009) acts as our sensitive dataset, as it is frequently used as a benchmark dataset, especially in the context of the differential privacy literature. CIFAR-10 is an MIT-licensed dataset consisting of 60,000 thumbnail-sized, colored images of 10 classes.

**Evaluation.** The model performance is delineated threefold: First, we evaluated a benchmark scenario in Fig. 3 (left) to compare our Secure Distributed DP-Helmet (cf. Section 4) to a DP-SGD-

---

[1]accessible at `https://github.com/google-research/simclr`, Apache-2.0 license

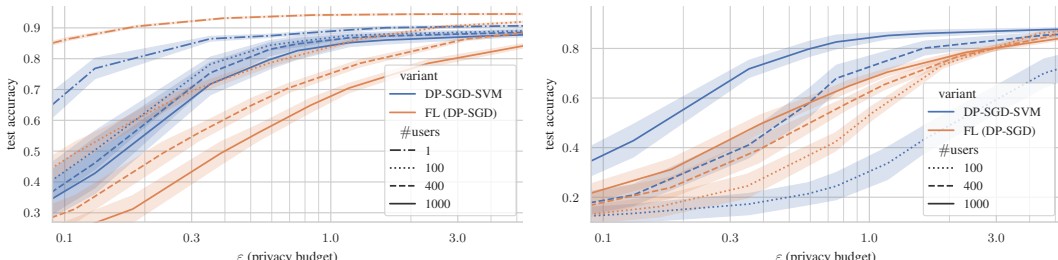

Figure 3: Classification accuracy compared to the privacy budget $\varepsilon$ (in $\log_{10}$-scale) of Secure Distributed DP-Helmet (cf. Section 4) and DP-SGD-based federated learning (FL) on CIFAR-10 dataset ($\delta = 10^{-5}$). (left) We use all available data points of CIFAR-10 for each line, spreading them among a differing number of users. (right) Different numbers of users with 50 data points per user.

based federated learning approach (DP-FL) on a single layer perceptron with softmax loss. There the approximately same number of data points is split across a various number of users ranging from 1 to 1000. Second, we also evaluated a realistic scenario in Fig. 3 (right) where we fixed the number of data points per user and report the performance increase obtained with more partaking users. Fig. 4 depicts the setting of Fig. 3 (right) for a fixed privacy budget. Third, we evaluated a scenario with truly many users as well as a local-DP setting in Fig. 2 where we rescale the privacy budget to accommodate the changed parameters.

The experiments lead to three conclusions: (RQ1) First, performance improves with an increasing number of users (cf. Fig. 3 (right)). Although the Secure Distributed DP-Helmet training performs subpar to DP-FL for few users, it takes off after about 400 users due to its vigorous performance gain with the number of users (cf. Fig. 4).

(RQ2) Second, in a scenario of a globally fixed number of data points (cf. Fig. 3 (left)) that are distributed over the users, Secure Distributed DP-Helmet's performance degrades more gracefully than that of DP-FL. Thm. 21 supports the more graceful decline; it states that averaging multiple of the here used SVM predictors eventually converges to the optimal SVM on all training data. The difference between 1 and 100 users is largely due to our assumption of $t = 50\%$ dishonest users, which means noise is scaled by a factor of $\sqrt{2}$. In comparison, DP-FL performs worse the more users $\mathcal{U}$ partake as the noise scales with $\mathcal{O}(|\mathcal{U}|^{1/2})$.

Figure 4: Classification accuracy versus #*users* with 50 data points per user for a fixed $\varepsilon = 0.5885$, $\delta = 10^{-5}$. Values for FL are interpolated.

(RQ3) Third, the advantage of our method over DP-FL becomes especially evident when considering significantly more users (cf. Fig. 2), such as is common in distributed training via smartphones. Here, DP-guarantees of $\varepsilon \leq 2 \cdot 10^{-4}$ become plausible with at least $87\%$ prediction performance for a task like CIFAR-10. Alternatively, leveraging Cor. 9 we can consider a local DP scenario (with $\Upsilon = 50$) without a trusted aggregator, yielding an accuracy of $84\%$ for $\varepsilon = 5 \cdot 10^{-4}$. Starting from $\Upsilon \geq 2$, a user-level sensitivity (cf. Cor. 10) is in the evaluated setting tighter than a data point dependent one; hence, the accuracy values are the same as for the local DP scenario.

(RQ4) We refer to Appx. F for an ablation study in the centrally trained setting for different learning algorithms than DP_SGD_SVM. In this setting, the here used DP_SGD_SVM has a worse privacy-utility trade-off than other DP learners like DP-SGD: for $\varepsilon = 0.59$, DP_SGD_SVM has an accuracy of $87.4\%$ while DP-SGD has $93.6\%$. The reasons include leakage via sequential composition (through DP-SGD-SVM's one-versus-rest multi-class approach) compared to DP-SGD's joint learning of all classes as well as its noise-correcting property from its iterative noise application.

**Computation costs.** For Secure Distributed DP-Helmet with $1,000$ users and a model size $l \approx 100,000$ for CIFAR-10, we need less than $0.2\,s$ for the client and $40\,s$ for the server, determined by extrapolating the experiments of (Bell et al., 2020, Table 2).

**Experimental setup.** Appx. D describes our experimental setup.

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

# A LIMITATIONS & DISCUSSION

**Distributional shifts between the public and sensitive datasets**    For pre-training our models, we leverage contrastive learning. While very effective generally, contrastive learning is susceptible to performance loss if the shape of the sensitive data used to train the SVMs is significantly different from the shape of the initial public training data.

**Multi-class classification**    As we train separate SVMs for the different classes, this approach works best if the number of classes is limited. Distributed DPHelmet can deal with multiple classes; CIFAR-10 has 10 classes. However, if a classification task has significantly more different classes, non-SVM-based approaches might perform better.

**Input Clipping**    DP_SGD_SVM requires a norm bound on the input data as it directly influences the SVM training. In many pre-training methods like SimCLR no natural bound exists thus we have to artificially norm clip the input data. To provide a non-data-dependent clipping bound in CIFAR-10 data, we determined the clipping bound on the CIFAR-100 dataset (here: $34.854$); its similar data distribution encompasses the output distribution of the pretraining reasonably well.

**Hyperparameter Search**    In SGD_SVM, we deploy two performance-crucial hyperparameters: the regularization weight $\Lambda$ as well as the predictor radius $R$, both of which influence noise scaling. In the noise scaling subterm $c/\Lambda + R$, the maximal predictor radius is naturally significantly smaller than $c/\Lambda$ due to the regularization penalty. Thus, an imperfect $R$ resulting from a non-hyperparameter-tuned SVM does not have a large impact on the performance. Estimating the regularization weight for a fixed $\varepsilon$ from public data is called hyperparameter freeness in prior work (Iyengar et al., 2019). For other $\varepsilon$ values we can fit a (linear) curve on a smaller but related public dataset (proposed by Chaudhuri et al. (2011)) or synthetic data (proposed by AMP-NT (Iyengar et al., 2019)) as smaller $\varepsilon$ prefer higher regularization weights and vice versa.

# B EXTENDED PRELIMINARIES

## B.1 DIFFERENTIAL PRIVACY

To ease our analysis, we consider a randomized mechanism $M$ to be a function translating a database to a random variable over possible outputs. Running the mechanism then is reduced to sampling from the random variable. With that in mind, the standard definition of differential privacy looks as follows.

**Definition 11** ($\approx_{\varepsilon,\delta}$ relation). *Let $Obs$ be a set of observations, and $\mathrm{RV}(Obs)$ be the set of random variables over $Obs$, and $\mathcal{D}$ be the set of all databases. A randomized algorithm $M : \mathcal{D} \to \mathrm{RV}(Obs)$ for a pair of datasets $\mathcal{D}, \mathcal{D}'$, we write $M(D) \approx_{\varepsilon,\delta} M(D')$ if for all tests $S \subseteq Obs$ we have*

$$\Pr[M(D) \in S] \leq \exp(\varepsilon) \Pr[M(D') \in S] + \delta. \tag{1}$$

**Definition 12** (Differential Privacy). *Let $Obs$ be a set of observations, and $\mathrm{RV}(Obs)$ be the set of random variables over $Obs$, and $\mathcal{D}$ be the set of all databases. A randomized algorithm $M : \mathcal{D} \to \mathrm{RV}(Obs)$ for all pairs of databases $\mathcal{D}, \mathcal{D}'$ that differ in at most 1 element is a $(\varepsilon, \delta)$-DP mechanism if we have*

$$M(D) \approx_{\varepsilon,\delta} M(D'). \tag{2}$$

In the context of machine learning, the randomized algorithm represents the training procedure of a predictor. Our distinguishing element is one data record of the database.

**Computational Differential Privacy**    Note that because of the secure summation, we technically require the computational version of differential privacy (Mironov et al., 2009), where the differential privacy guarantees are defined against computationally bounded attackers; the resulting increase in $\delta$ is negligible and arguments about computationally bounded attackers are omitted to simplify readability.

**Definition 13** (Computational $\approx_{\varepsilon,\delta}^c$ Differential Privacy). *Let $\mathcal{D}$ be the set of all databases and $\eta$ a security parameter. A randomized algorithm $M : \mathcal{D} \to \mathrm{RV}(Obs)$ for a pair of datasets $\mathcal{D}, \mathcal{D}'$, we write $M(D) \approx_{\varepsilon,\delta}^c M(D')$ if for any polynomial-time probabilistic attacker*

$$\Pr[A(M(D)) = 0] \leq \exp(\varepsilon) \Pr[A(M(D')) = 1] + \delta(\eta). \tag{3}$$

*For all pairs of databases $\mathcal{D}, \mathcal{D}'$ that differ in at most $1$ element $M$ is a computational $(\varepsilon, \delta(\eta))$-DP mechanism if we have*

$$M(D) \approx_{\varepsilon,\delta}^c M(D'). \tag{4}$$

## B.2 SECURE SUMMATION

Hiding intermediary local training results as well as ensuring their integrity is provided by an instance of secure multi-party computation (SMPC) called secure summation (Bonawitz et al., 2017; Bell et al., 2020). It is targeted to comply with distributed summations across a huge number of parties. In fact, Bell et al. (2020) has a computational complexity for $n$ users on an $l$-sized input of $\mathcal{O}(\log^2 n + l \log n)$ for the client and $\mathcal{O}(n(\log^2 n + l \log n))$ for the server as well as a communication complexity of $\mathcal{O}(\log n + l)$ for the client and $\mathcal{O}(n(\log n + l))$ for the server thus enabling an efficient run-through of roughly $10^9$ users without biasing towards computationally equipped users. Additionally, it offers resilience against client dropouts and colluding adversaries, both of which are substantial features for our distributed setting:

**Theorem 14** (Secure Aggregation $\pi_{SecAgg}$ in the semi-honest setting exists (Bell et al., 2020)). *Let $s_1, \ldots, s_n$ be the $d$-dimensional inputs of the clients $U^{(1)}, \ldots, U^{(n)}$. Let $\mathcal{F}$ be the ideal secure summation function: $\mathcal{F}(s_1, \ldots, s_n) := 1/n \sum_{i=1}^n s_i$. If secure authentication encryption schemes and authenticated key agreement protocol exist, the fraction of dropouts (i.e., clients that abort the protocol) is at most $\rho \in [0, 1]$, at most a $\gamma \in [0, 1]$ fraction of clients is corrupted ($C \subseteq \{U^{(1)}, \ldots, U^{(n)}\}, |C| = \gamma n$), and the aggregator is honest-but-curious, there is a secure summation protocol $\pi_{SecAgg}$ for a central aggregator and $n$ clients that securely emulates $\mathcal{F}$ in the following sense: there is a probabilistic polynomial-time simulator $Sim_{\mathcal{F}}$ such that $Real_{\pi_{SecAgg}}(s_1, \ldots, s_n)$ is statistically indistinguishable from $Sim_{\mathcal{F}}(C, \mathcal{F}(s_1, \ldots, s_n))$, i.e., for an unbounded attacker $\mathcal{A}$ there is a negligible function $\nu$ such that*

*Advantage$(\mathcal{A}) =$*
$$|\Pr[\mathcal{A}(Real_{\pi_{SecAgg}}(s_1, \ldots, s_n)) = 1] - \Pr[\mathcal{A}(Sim_{\mathcal{F}}(C, \mathcal{F}(s_1, \ldots, s_n))) = 1]| \leq \nu(\eta).$$

## B.3 PRE-TRAINING TO BOOST DP PERFORMANCE

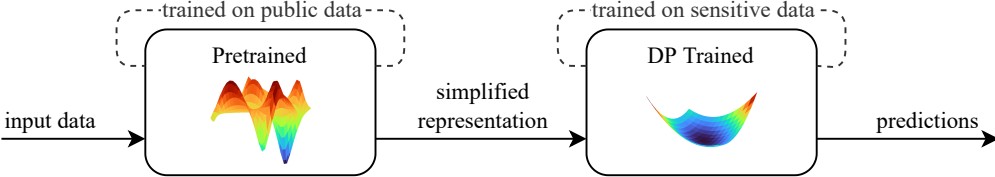

Figure 5: Pre-training: Schematic overview. Dashed lines denote data flow in the training phase and solid lines in the inference phase.

Recent work (Tramèr & Boneh, 2021; De et al., 2022) has shown that strong feature extractors (such as SimCLR (Chen et al., 2020a;b)), trained in an unsupervised manner, can be combined with simple learners to achieve strong utility-privacy tradeoffs for high-dimensional data sources like images. As a variation to transfer learning, it delineates a two-step process (cf. Fig. 5), where a simplified representation of the high-dimensional data is learned first before a tight privacy algorithm like DP_SGD_SVM conducts the prediction process on these simplified representations. For that, two data sources are compulsory: a public data source which is used to undertake the learning of a framework aimed to obtain pertinent simplified representations in addition to our sensitive data source that conducts the prediction process in a differentially private manner. Thereby the sensitive dataset is protected while strong expressiveness is assured through the use of the feature reduction network. Also note that a homogeneous data distribution of the public and the sensitive data is not necessarily required.

Recent work has shown that for several applications such representation reduction frameworks can be found, such as SimCLR for pictures, FaceNet for face images, UNet for segmentation, or GPT-3 for language data. Without loss of generality, we focus in this work on the unsupervised SimCLR feature

reduction network (Chen et al., 2020a;b). SimCLR uses contrastive loss and image transformations to align the embeddings of similar images while keeping those of dissimilar images separate (Chen et al., 2020a). It is based upon a self-supervised training scheme called contrastive loss where no labeled data is required. Labelless data is especially useful as it exhibits possibilities to include large-scale datasets which would otherwise be unattainable due to the labeling efforts needed.

### B.4 DP_SGD_SVM

**Definition 15.** *The Huber loss according to Chaudhuri et al. (2011, Equation 7) is with a smoothness parameter $h$ defined as*

$$\ell_{huber}(z) := \begin{cases} 0 & \text{if } z > 1 + h \\ \frac{1}{4h}(1 + h - z)^2 & \text{if } |1 - z| \leq h \\ 1 - z & \text{if } z < 1 - h \end{cases}.$$

## C   RELATED WORK

### C.1   PRIVACY-PRESERVING DISTRIBUTED MACHINE LEARNING

There is a rich body of literature about different differentially private distributed learning techniques that protect any individual data point (sometimes called distributed learning with global DP guarantees). One direction uses an untrusted central aggregator; users locally add noise to avoid leakage toward the aggregator. This method computationally scales well with the number of users. Another direction utilizes cryptographic protocols to jointly train a model without a central aggregator. This direction requires less noise for privacy, but the cryptographic protocols face scalability challenges.

For local noising, the most prominent and flexible approach is federated learning (McMahan et al., 2017) with DP-SGD approximation (Abadi et al., 2016) (DP-FL). DP-FL proposes each of the $n$ users locally train with the DP-SGD algorithm and share their local gradient updates with a central aggregator. This aggregator updates a global model with the average of the noisy local updates, leading to noise overhead in the order of $\sqrt{n}$.

This noise overhead can be completely avoided by PPDML protocols that rely on cryptographic methods to hide intermediary training updates from a central aggregator. There are several secure distributed learning methods that protect the contributions during training but do not come with privacy guarantees for the model such as DP: an attacker (e.g., a curious training party) can potentially extract information about the training data from the model. As we focus on differentially private distributed learning methods (PPDML in this paper), we will neglect those methods.

cpSGD (Agarwal et al., 2018) is a PPDML protocol that utilizes SMPC methods to honestly generate noise and compute DP-SGD. While cpSGD provides the full flexibility of SGD, it does not scale to millions of users as it relies on expensive SMPC methods. Another recent PPDML work (Truex et al., 2019) relies on a combination of SMPC and DP methods. This work, however, also does not scale to millions of users.

Another line of research aims for the stronger privacy goal of protecting a user's entire input (called local DP) during distributed learning (Balle et al., 2020b; Girgis et al., 2021). Due to the strong privacy goal, federated learning with local DP tends to achieve weaker accuracy. With Cor. 9, evaluated in Fig. 2 in Section 5, we show how Secure Distributed DP-Helmet achieves a comparable guarantee via group privacy: given enough users, any user can protect their entire dataset at once while we still reach good accuracy.

For DP training of SVMs, there exist other methods, such as objective perturbation and gradient perturbation. When performed under SMPC-based distributed training, both methods would require a significantly higher number of SMPC invocations; hence, they are unsuited for the goals of this work. Appx. C.2 discusses those approaches in detail.

### C.2   DIFFERENTIALLY PRIVATE EMPIRICAL RISK MINIMIZATION

On differentially private empirical risk minimization for convex loss functions (Chaudhuri et al., 2011), which is utilized in this work, the literature discusses three directions: output perturbation,

objective perturbation, and gradient perturbation. Output perturbation (Chaudhuri et al., 2011; Wu et al., 2017) estimates a sensitivity on the final model without adding noise, and only in the end adds noise that is calibrated to this sensitivity. We rely on output perturbation because it enables us to only have a single invocation of an SMPC protocol at the end to merge the models while still achieving the same low sensitivity as if the model was trained at a trustworthy central party that collects all data points, trains a model and adds noise in the end.

Objective perturbation (Chaudhuri et al., 2011; Kifer et al., 2012; Iyengar et al., 2019; Bassily et al., 2019) adds noise to the objective function instead of adding noise to the final model. In principle, SMPC could also be used to emulate the situation that a central party as above trains a model via objective perturbation. Yet, in that case, each party would have to synchronize with every other party far more often, as no party would be allowed to learn how exactly the objective function would be perturbed. That would result in far higher communication requirements.

Concerning gradient perturbation (Bassily et al., 2014; Wang et al., 2017; Feldman et al., 2018; Bassily et al., 2019; Feldman et al., 2020), recent work has shown tight privacy bounds. In order to achieve the same low degree of required noise as in a central setting, SMPC could be utilized. Yet, for SGD also multiple rounds of communication would be needed as the privacy proof (for convex optimization) does not take into account that intermediary gradients are leaked. Hence, the entire differentially private SGD algorithm for convex optimization would have to be computed in SMPC, similar to cpSGD (see above).

## D EXPERIMENTAL SETUP

We leveraged 5-repeated 6-fold stratified cross-validation for all experiments unless stated differently. Privacy Accounting has been undertaken either by using the privacy bucket (Meiser & Mohammadi, 2018; Sommer et al., 2019) toolbox[2] or, for Gaussians without subsampling, with Sommer et al. (2019, Theorem 5) where both can be extended to multivariate Gaussians (see Appx. L). We note that with either of these tactics, $\varepsilon \in \mathcal{O}(|K|^{1/2})$. The $\delta$ parameter of differential privacy has been set to $\delta = 10^{-5}$ if not stated otherwise, which is for the CIFAR-10 dataset always below $1/n$, where $n$ is the sum of the size of all local datasets.

Concerning computation resources, for our experiments, we trained 1000 DP_SGD_SVM with 50 data points each, which took 10 minutes on a machine with 2x *Intel Xeon Platinum 8168*, 24 Cores @2.7 GHz with an Nvidia A100 and allocated 16GB RAM.

For DP_SGD_SVM-based experiments, we utilize the strongly convex projected stochastic gradient descent algorithm (PSGD) as used by Wu et al. (2017). More specifically, we chose a batch size of 20, the Huber loss with a smoothness parameter $h = 0.1$, a hypothesis space radius $R \in \{0.04, 0.05, 0.06, 0.07, 0.08\}$, a regularization parameter $\Lambda \in \{10, 100, 200\}$, and trained for 500 epochs; for the variant where we protect the whole local dataset, we have chosen a different $\Lambda \in \{0.5, 1, 2, 5\}$ and $R \in \{0.06, 0.07\}$.

In every experiment, we chose for each parameter combination the best performing regularization parameter $\Lambda$ as well as $R$, i.e. those values that lead to the best mean accuracy. This is highly important, as the regularization parameter not only steers the utility but also the amount of noise needed to the effect where there is a sweet spot for each noise level where the amount of added noise is on the edge of still being bearable.

For the federated learning experiments, we utilized the *opacus*[3] PyTorch library (Yousefpour et al., 2021), which implements DP-SGD (Abadi et al., 2016). We loosely adapted our hyperparameters to the ones reported by Tramèr & Boneh (2021) who already evaluated DP-SGD on SimCLR's embeddings for the CIFAR-10 dataset. In detail, the neural network is a single-layer perceptron with 61 450 trainable parameters on a 6 144-d input and $10d$ output. The loss function is the categorical cross-entropy on a softmax activation function and training has been performed with stochastic gradient descent. Furthermore, we set the learning rate to 4, the Poisson sample rate $q := 1024/50000$ which in expectation samples a batch size of 1024, trained for 40 epochs, and norm-clipped the gradients with a clipping bound $c := 0.1$.

---

[2]accessible at `https://github.com/sommerda/privacybuckets`, MIT license
[3]accessible at `https://github.com/pytorch/opacus/`, Apache-2.0 license

In the distributed training scenario, instead of running an end-to-end experiment with full SMPC clients, we evaluate a functionally equivalent abstraction without cryptographic overhead. In our experiments, we randomly split the available data points among the users and emulated scenarios where not all data points were needed by taking the first training data points. However, the validation size remained constant. Moreover, for DP-SGD-based federated learning, we kept a constant batch size whenever enough data is available i.e. increased the sampling rate as follows: $q' := {}^{1024}/_{20000}$ for 20000, $q'' := {}^{1024}/_{5000}$ for 5000, and $q'' := {}^{1023}/_{1024}$ for 500 available data points ($|\mathcal{U}| \cdot N$). For DP-SGD-based FL, we emulated a higher number of users by dividing the noise multiplier $\sigma$ by $|\mathcal{U}|^{1/2}$ to the benefit of DP-FL. The justification for dividing by $|\mathcal{U}|^{1/2}$ is that in FL the model performance is not expected to differ as the mean of the gradients of one user is the same as the mean of gradients from different users: SGD computes, just as FL, the mean of the gradients. Yet, the noise will increase by a factor of $|\mathcal{U}|^{1/2}$. Hence, we optimistically assume that everything stays the same, just the noise increases by a factor of $|\mathcal{U}|^{1/2}$.

# E PRE-TRAINING VISUALISATION

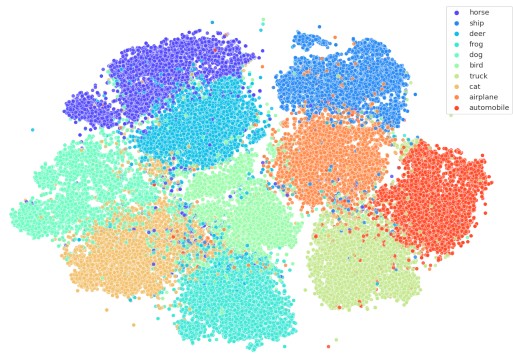

Figure 6: 2-d projection of the CIFAR-10 dataset via t-SNE (Van der Maaten & Hinton, 2008) with colored labels. Note that t-SNE is defined on the local neighborhood thus global patterns or structures may be arbitrary.

# F EXTENDED ABLATION STUDY (CENTRALIZED SETTING)

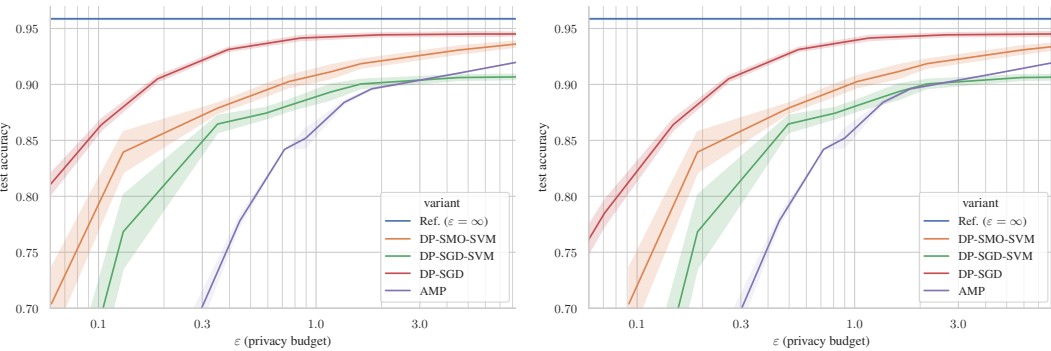

Figure 7: Classification accuracy compared to the privacy budget $\varepsilon$ of DP_SGD_SVM (cf. Section 3.1), DP_SMO_SVM where only the optima are perturbed, DP-SGD (1-layer perceptron) (Abadi et al., 2016), and AMP (SVM with objective perturbation) (Iyengar et al., 2019) on CIFAR-10 benchmark dataset (left: $\delta = 10^{-5}$, right: $\delta = 2 \cdot 10^{-8} \ll {}^{1}/_{dataset\_size}$). For comparison, we report a non-private SVM baseline.

### F.1 SETUP OF THE ABLATION STUDY

For DP_SMO_SVM-based experiments, we used the *liblinear* (Fan et al., 2008) library via the Scikit-Learn method *LinearSVC*[4] for classification. *Liblinear* is a fast C++ implementation that uses the SVM-agnostic sequential minimal optimization (SMO) procedure. However, it does not offer a guaranteed and private convergence bound.

More specifically, we used the $L_2$-regularized hinge loss, an SMO convergence tolerance of $tol := 2 \cdot 10^{-12}$ with a maximum of $10{,}000$ iterations which were seldom reached, and a logarithmically spaced inverse regularization parameter $C \in \{\{3, 6\} \cdot 10^{-8}, \{1, 2, 3, 6\} \cdot 10^{-7}, \{1, 2, 3, 6\} \cdot 10^{-6}, \{1, 2, 3, 6\} \cdot 10^{-5}, \{1, 2\} \cdot 10^{-4}\}$. To better fit with the *LinearSVC* implementation, the original loss function is rescaled by $1/\Lambda$ and $C$ is set to $1/\Lambda \cdot n$ with $n$ as the number of data points. Furthermore, for distributed DP_SMO_SVM training we extended the range of the hyperparameter $C$ – whenever appropriate – up to $3 \cdot 10^{-3}$ which becomes relevant in a scenario with many users and few data points per user. Similar to DP_SGD_SVM-based experiment, the best performing regularization parameter $C$ was selected for each parameter combination.

The non-private reference baseline uses a linear SVM optimized via SMO with the hinge loss and an inverse regularization parameter $C = 2$ (best performing of $C \in \{\leq 5 \cdot 10^{-5}, 0.5, 1, 2\}$).

For the ablation study, we also included the Approximate Minima Perturbation (AMP) algorithm[5] (Iyengar et al., 2019) which resembles an instance of objective perturbation. There, we used a (80–20)-train-test split with 10 repeats and the following hyperparameters: $L \in \{0.1, 1.0, 34.854\}$, eps_frac $\in \{.9, .95, .98, .99\}$, eps_out_frac $\in \{.001, .01, .1, .5\}$. We selected ($L = 1, eps\_out\_frac = 0.001$, $eps\_frac = 0.99$) as a good performing parameter combination for AMP. For better performance, we resembled the GPU-capable *bfgs_minimize* from the Tensorflow Probability package. To provide better privacy guarantees, we leveraged the results of Kairouz et al. (2015a); Murtagh & Vadhan (2016) for tighter composition bounds on arbitrary DP mechanisms.

### F.2 RESULTS OF THE ABLATION STUDY

For the extended ablation study, we considered the centralized setting (only 1 user) and compare different algorithms as well as different values for the privacy parameter $\delta$. The results are depicted in Fig. 7 and display four algorithms: firstly, the differentially private Support Vector Machine with SGD-based training DP_SGD_SVM (cf. Section 3.1), secondly, a similar differentially private SVM but with SMO-based training which does not offer a guaranteed and private convergence bound, thirdly, differentially private Stochastic Gradient descent (DP-SGD) (Abadi et al., 2016) applied on a 1-layer perceptron with the cross-entropy loss, and fourthly, approximate minima perturbation (AMP) (Iyengar et al., 2019) which is based upon an SVM with objective perturbation. Note that, only DP_SMO_SVM and DP_SGD_SVM have an output sensitivity and are thus suited for this efficient Secure Distributed DP-Helmet scheme.

While all algorithms come close to the non-private baseline with rising privacy budgets $\varepsilon$, we observe that although DP-SGD performs best, DP_SMO_SVM comes considerably close, DP_SGD_SVM has a disadvantage above DP_SMO_SVM of about a factor of 2, and AMP a disadvantage of about a factor of 4. We suspect that DP-SGD is able to outperform the other variants as it is the only contestant which directly optimizes for the multi-class objective via the cross-entropy loss while others are only able to simulate it via the one-vs-rest (ovr) SVM training scheme. Although DP_SMO_SVM renders best of the variants with an output sensitivity, it does not offer a privacy guarantee when convergence is not reached. In the case of AMP, we have an inherent disadvantage of about a factor of 3 due to an unknown output distribution, and thus bad composition results in the multi-class SVM. Here, the privacy budget of AMP roughly scales linearly with the number of classes.

For DP-SGD, DP_SGD_SVM, and DP_SMO_SVM, Fig. 7 shows that a smaller and considerably more secure privacy parameter $\delta \ll 1/dataset\_size$ is supported although reflecting on the reported privacy budget $\varepsilon$.

---

[4]`https://scikit-learn.org/stable/modules/generated/sklearn.svm.LinearSVC.html`, BSD-3-Clause license

[5]reference implementation by the authors: `https://github.com/sunblaze-ucb/dpml-benchmark`, MIT license

## G    PROOF OF COR. 6

We recall Cor. 6:

**Corollary 6** (Sensitivity of Secure Distributed DP-Helmet). *Given a configuration $\zeta$, Secure Distributed DP-Helmet($\zeta$) (cf. Alg. 2) without adding noise, i.e. $\mathrm{avg}(T(\mho))$, has a sensitivity of $s \cdot 1/|\mathcal{U}|$ for each class $k \in K$.*

*Proof.* Without loss of generality, we consider one arbitrary class $k \in K$. We know that $T$ is an $s$-sensitivity bounded algorithm thus

$$s = \max_{D_0^{(i)} \sim D_1^{(i)}} \left| T(D_0^{(i)}, \xi, k) - T(D_1^{(i)}, \xi, k) \right| \tag{5}$$

with $D_0^{(i)}$ and $D_1^{(i)}$ as 1-neighboring datasets. For instance, for $T = \mathrm{SGD\_SVM}$ we have $s = \frac{2(c+R\Lambda)}{N\Lambda}$ (cf. Lem. 2).

By Alg. 2, we take the average of multiple local models, i.e. $\mathrm{avg}(T(\mho)) = \frac{1}{|\mathcal{U}|} \sum_{i=1}^{|\mathcal{U}|} T(D^{(i)}, \xi, K)$. The challenge element – i.e. the element that differs between $D_0^{(i)}$ and $D_1^{(i)}$ – is only contained in one of the $|\mathcal{U}|$ SGD_SVM's. By the application of the parallel composition theorem, we know that the sensitivity reduces to

$$\max_{D_0^{(i)} \sim D_1^{(i)}, \forall i=0,\dots,|\mathcal{U}|} \left| \frac{1}{|\mathcal{U}|} \sum_{i=1}^{|\mathcal{U}|} T(D_0^{(i)}, \xi, k) - \frac{1}{|\mathcal{U}|} \sum_{i=1}^{|\mathcal{U}|} T(D_1^{(i)}, \xi, k) \right| = s \cdot \frac{1}{|\mathcal{U}|}.$$

Hence, the constant $1/|\mathcal{U}|$ factor reduces the sensitivity by a factor of $1/|\mathcal{U}|$.    $\square$

## H    PROOF OF LEM. 7

We recall Lem. 7:

**Lemma 7.** *Given a configuration $\zeta$ and any noise scale $\tilde{\sigma}$, then $\frac{1}{|\mathcal{U}|} \sum_{i=1}^{|\mathcal{U}|} \mathcal{N}(0, (\tilde{\sigma} \cdot 1/\sqrt{|\mathcal{U}|})^2) = \mathcal{N}(0, (\tilde{\sigma} \cdot 1/|\mathcal{U}|)^2)$.*

*Proof.* We have to show that

$$\tfrac{1}{|U|} \sum_{i=1}^{|U|} \mathcal{N}(0, (\tilde{\sigma} \cdot \tfrac{1}{\sqrt{|\mathcal{U}|}})^2) = \mathcal{N}(0, (\tilde{\sigma} \cdot \tfrac{1}{|\mathcal{U}|})^2). \tag{6}$$

It can be shown that the sum of normally distributed random variables behaves as follows: Let $X \sim \mathcal{N}(\mu_X, \sigma_X^2)$ and $Y \sim \mathcal{N}(\mu_Y, \sigma_Y^2)$ two independent normally-distributed random variables, then their sum $Z = X + Y$ equals $Z \sim \mathcal{N}(\mu_X + \mu_Y, \sigma_X^2 + \sigma_Y^2)$ in the expectation.

Thus, in this case, we have

$$\tfrac{1}{|U|} \sum_{i=1}^{|U|} \mathcal{N}(0, (\tilde{\sigma} \cdot \tfrac{1}{\sqrt{|\mathcal{U}|}})^2) = \tfrac{1}{|U|} \mathcal{N}(0, |U| \cdot (\tilde{\sigma} \cdot 1/\sqrt{|\mathcal{U}|})^2) = \tfrac{1}{|U|} \mathcal{N}(0, \tilde{\sigma}^2).$$

As the normal distribution belongs to the location-scale family, we get $\mathcal{N}(0, (\tilde{\sigma} \cdot 1/|\mathcal{U}|)^2)$.    $\square$

## I    PROOF OF THM. 8

We state the full version of Thm. 8:

**Theorem 8** (Main Theorem, full). *Given a configuration $\zeta$, a maximum fraction of dropouts $\rho \in [0, 1]$, and a maximum fraction of corrupted clients $\gamma \in [0, 1]$, if secure authentication encryption schemes and authenticated key agreement protocol exist, then Secure Distributed DP-Helmet($\zeta$) (cf. Alg. 2) satisfies computational $(\varepsilon, \delta + \nu_1)$-DP with $\varepsilon \geq \sqrt{2 \ln 1.25/(\delta/|K|)} \cdot |K| \cdot 1/\sigma$, for $\nu_1 := (1 + \exp(\varepsilon)) \cdot \nu(\eta)$ and a function $\nu$ negligible in the security parameter $\eta$ used in secure aggregation.*

*Proof.* We first show $(\varepsilon, \delta)$-DP for a variant $M_1$ of Secure Distributed DP-Helmet that uses the ideal summation protocol $\mathcal{F}$ instead of $\pi_{SecAgg}$. Then, we conclude from Thm. 14 that for Secure Distributed DP-Helmet (abbreviated as $M_2$) which uses the real secure summation protocol $\pi_{SecAgg}$ for some negligible function $\nu_1$ $(\varepsilon, \delta + \nu_1)$-DP holds.

Recall that we assume at least $t \cdot |\mathcal{U}|$ many honest users. As we solely rely on the honest $t \cdot |\mathcal{U}|$ to contribute correctly distributed noise to the learning algorithm $T$, we have for each class similar to Lem. 7

$$\frac{1}{|\mathcal{U}|} \sum_{i=1}^{t \cdot |\mathcal{U}|} \mathcal{N}(0, (\tilde{\sigma} \cdot \frac{1}{\sqrt{|\mathcal{U}|}})^2) = \sum_{i=1}^{t \cdot |\mathcal{U}|} \mathcal{N}(0, (\tilde{\sigma} \cdot \frac{1}{|\mathcal{U}|\sqrt{|\mathcal{U}|}})^2)$$

$$= \mathcal{N}(0, (\tilde{\sigma} \cdot \frac{\sqrt{t \cdot |\mathcal{U}|}}{|\mathcal{U}|\sqrt{|\mathcal{U}|}})^2) = \mathcal{N}(0, (\tilde{\sigma} \cdot \frac{\sqrt{t}}{|\mathcal{U}|})^2).$$

Hence, we scale the noise parameter $\tilde{\sigma}$ with $1/\sqrt{t}$ and get

$$\frac{1}{|\mathcal{U}|} \sum_{i=1}^{t|\mathcal{U}|} \mathcal{N}(0, (\tilde{\sigma} \cdot \frac{1}{\sqrt{t}} \cdot \frac{1}{\sqrt{|\mathcal{U}|}})^2) = \mathcal{N}(0, (\tilde{\sigma} \cdot \frac{1}{|\mathcal{U}|})^2).$$

By Cor. 6, Lem. 7, and Lem. 3, we know that $M_1$ satisfies $(\varepsilon, \delta)$-DP (with the parameters as described above).

Hence, considering an unbounded attacker $\mathcal{A}$ and Thm. 14, we know that for any pair of neighboring data sets $D, D'$ the following holds

$$\Pr[\mathcal{A}(\mathcal{M}_1(D)) = 1] \leq \exp(\varepsilon) \Pr[\mathcal{A}(\mathcal{M}_1(D')) = 1] + \delta$$

By Thm. 14, we know that $\pi_{SecAgg}(s_1, \ldots, s_n)$ securely emulates $\mathcal{F}$ (w.r.t. an unbounded attacker). Hence, there is a negligible function $\nu$ such that for any neighboring data sets $D, D'$ (differing in at most one element) the following holds w.l.o.g.:

$$\Pr[\mathcal{A}(\mathcal{M}_2(D)) = 1] - \nu(\eta) \leq \Pr[\mathcal{A}(Sim_{\mathcal{F}}(\mathcal{M}_1(D))) = 1]. \tag{7}$$

For the attacker $\mathcal{A}'$ that first applies $Sim$ and then $\mathcal{A}$, we get:

$$\Pr[\mathcal{A}(\mathcal{M}_2(D)) = 1] - \nu(\eta) \leq \exp(\varepsilon) \Pr[\mathcal{A}(Sim_{\mathcal{F}}(\mathcal{M}_1(D'))) = 1] + \delta \tag{8}$$

$$\leq \exp(\varepsilon) (\Pr[\mathcal{A}(\mathcal{M}_2(D')) = 1] + \nu(\eta)) + \delta \tag{9}$$

thus we have

$$\Pr[\mathcal{A}(\mathcal{M}_2(D)) = 1] \leq \exp(\varepsilon) \Pr[\mathcal{A}(\mathcal{M}_2(D')) = 1] + \delta + (1 + \exp(\varepsilon)) \cdot \nu(\eta). \tag{10}$$

From a similar argumentation it follows that

$$\Pr[\mathcal{A}(\mathcal{M}_2(D')) = 1] \leq \exp(\varepsilon) \Pr[\mathcal{A}(\mathcal{M}_2(D)) = 1] + \delta + (1 + \exp(\varepsilon)) \cdot \nu(\eta) \tag{11}$$

holds.

Hence, with $\nu_1 := (1 + \exp(\varepsilon)) \cdot \nu(\eta)$ the mechanism Secure Distributed DP-Helmet mechanism $\mathcal{M}_2$ which uses $\pi_{SecAgg}$ is $(\varepsilon, \delta + \nu_1)$-DP. As $\nu$ is negligible and $\varepsilon$ is constant, $\nu_1$ is negligible as well. $\qquad\square$

## J  PROOF OF COR. 10

We recall Cor. 10:

**Corollary 10** (User-level sensitivity). *Given a learning algorithm $T$, we say that $T$ is $R$-norm bounded if for any input dataset $D$ with $N = |D|$, any hyperparameter $\xi$, and all classes $k \in K$, $\|T(D, \xi, k)\| \leq R$. Any $R$-norm bounded learning algorithm $T$ has a sensitivity $s = 2R$. In particular, $T + \mathcal{N}(0, (\sigma \cdot s \cdot I_d)^2)$ satisfies $(\varepsilon, \delta)$, $\Upsilon$-group differential privacy with $\Upsilon = N$ and $\varepsilon \geq \sqrt{2 \ln 1.25/(\delta/|K|)} \cdot |K| \cdot 1/\sigma$, where $\mathcal{N}(0, (\sigma \cdot s \cdot I_d)^2)$ is spherical multivariate Gaussian noise and $\sigma$ a noise multiplier.*

*Proof.* We know that the sensitivity of the learning algorithm $T$ is defined as $s = \max_{D \sim D'} \|T(D, \xi, k) - T(D', \xi, k)\|$ for $\Upsilon$-neighboring datasets $D, D'$. Thus, in our case we have $s = 2R$ since any $T(\_, \xi, k) \in [-R, R]$. As this holds independent on the dataset and by Lem. 3 and by Lem. 16, we can protect any arbitrary number of data points per user, i.e. we have $\Upsilon$-group DP. $\square$

## K   GROUP PRIVACY REDUCTION OF MULTIVARIATE GAUSSIAN

**Lemma 16.** *Let* $\mathrm{pdf}_{\mathcal{N}(A,B)}[x]$ *denote the probability density function of the multivariate Gaussian distribution with location and scale parameters* $A, B$ *which is evaluated on an atomic event* $x$. *For any atomic event* $x$, *any covariance matrix* $\Sigma$, *any group size* $k \in \mathbb{N}$, *and any mean* $\mu$, *we get*

$$\frac{\mathrm{pdf}_{\mathcal{N}(0,k^2\Sigma)}[x]}{\mathrm{pdf}_{\mathcal{N}(\mu,k^2\Sigma)}[x]} = \frac{\mathrm{pdf}_{\mathcal{N}(0,\Sigma)}[x/k]}{\mathrm{pdf}_{\mathcal{N}(\mu/k,\Sigma)}[x/k]}. \tag{12}$$

*Proof.*

$$\frac{\mathrm{pdf}_{\mathcal{N}(0,k^2\Sigma)}[x]}{\mathrm{pdf}_{\mathcal{N}(\mu,k^2\Sigma)}[x]} = \frac{\frac{1}{det(2\pi k^2\Sigma)}\exp(-\frac{1}{2}x^T k^2\Sigma^{-1}x)}{\frac{1}{det(2\pi k^2\Sigma)}\exp(-\frac{1}{2}\underbrace{(x-\mu)^T k^2\Sigma^{-1}(x-\mu)}_{=x^T k^2\Sigma^{-1}x - \mu^T k^2\Sigma^{-1}x - x^T k^2\Sigma^{-1}\mu + \mu^T k^2\Sigma^{-1}\mu})} \tag{13}$$

$$= \exp(-\frac{1}{2}(-\mu^T k^2\Sigma^{-1}x - x^T k^2\Sigma^{-1}\mu + \mu^T k^2\Sigma^{-1}\mu)) \tag{14}$$

$$= \exp(-\frac{1}{2}k^2 \cdot (-\mu^T\Sigma^{-1}x - x^T\Sigma^{-1}\mu + \mu^T\Sigma^{-1}\mu)) \tag{15}$$

for $\mu_1 := \mu/k$

$$= \exp(-\frac{1}{2} \cdot k(-\mu_1^T\Sigma^{-1}x - x^T\Sigma^{-1}\mu_1 + \mu_1^T\Sigma^{-1}\mu_1/k)) \tag{16}$$

for $x_1 := x/k$

$$= \exp(-\frac{1}{2} \cdot (-\mu_1^T\Sigma^{-1}x_1 - x_1^T\Sigma^{-1}\mu_1 + \mu_1^T\Sigma^{-1}\mu_1)) \tag{17}$$

$$= \exp(-\frac{1}{2} \cdot (-\mu_1^T\Sigma^{-1}x_1 - x_1^T\Sigma^{-1}\mu_1 + \mu_1^T\Sigma^{-1}\mu_1)) \tag{18}$$

$$= \frac{\frac{1}{det(2\pi\Sigma)}\exp(-\frac{1}{2}x_1^T\Sigma^{-1}x_1)}{\frac{1}{det(2\pi\Sigma)}\exp(-\frac{1}{2}(x_1-\mu_1)^T k^2\Sigma^{-1}(x_1-\mu_1))} \tag{19}$$

$$= \frac{\mathrm{pdf}_{\mathcal{N}(0,\Sigma)}[x/k]}{\mathrm{pdf}_{\mathcal{N}(\mu/k,\Sigma)}[x/k]} \tag{20}$$

$\square$

As the Gaussian distribution belongs to the location-scale family, Lem. 16 directly implies that the $(\varepsilon, \delta)$-DP guarantees of using $\mathcal{N}(0, k^2 \Sigma)$ noise for sensitivity $k$ and using $\mathcal{N}(0, \Sigma)$ for sensitivity 1 are the same.

## L   REPRESENTING MULTIVARIATE GAUSSIANS AS UNIVARIATE GAUSSIANS

For the sake of completeness, we rephrase a proof that we first saw in Abadi et al. (2016) that argues that sometimes the multivariate Gaussian mechanism can be reduced to the univariate Gaussian mechanism.

**Lemma 17.** *Let* $\mathrm{pdf}_{\mathcal{N}(\mu,\mathrm{diag}(\sigma^2))}$ *denote the probability density function of a multivariate* $(p \geq 1)$ *spherical Gaussian distribution with location and scale parameters* $\mu \in \mathbb{R}^p, \sigma \in \mathbb{R}^p_+$. *Let* $M_{gauss,p,q}$ *be the* $p$ *dimensional Gaussian mechanism* $D \mapsto q(D) + \mathcal{N}(0, \sigma^2 \cdot I_p)$ *for* $\sigma^2 > 0$ *of a function* $q : \mathcal{D} \to \mathbb{R}^p$, *where* $\mathcal{D}$ *is the set of datasets. Then, for any* $p \geq 1$, *if* $q$ *is* $s$-sensitivity-bounded, *then for any* $p \geq 1$, *there is another* $s$-sensitivity-bounded *function* $q' : \mathcal{D} \to \mathbb{R}$ *such that the following holds: for all* $\varepsilon \geq 0, \delta \in [0, 1]$ *if* $M_{gauss,1,q'}$ *satisfies* $(\varepsilon, \delta)$-ADP, *then* $M_{gauss,p,q}$ *satisfies* $(\varepsilon, \delta)$-ADP.

*Proof.* First observe that for any $s$-sensitivity-bounded function $q''$, two adjacent inputs $D, D'$ (differing in one element) with $\|q''(D) - q''(D')\|_2 = s$ are worst-case inputs. As a spherical Gaussian distribution (covariance matrix $\Sigma = \sigma^2 \cdot I_{p \times n}$) is rotation invariant, there is a rotation such that the difference only occurs in one dimension and has length $s$. Hence, it suffices to analyze a univariate Gaussian distribution with sensitivity $s$. Hence, the privacy loss distribution of both mechanisms (for the worst-case inputs) is the same. As a result, for all $\varepsilon \geq 0, \delta \in [0, 1]$ (i.e. the privacy profile is the same) if $(\varepsilon, \delta)$-ADP holds for the univariate Gaussian mechanism it also holds for the multivariate Gaussian mechanism. $\qquad\square$

## M   STABILITY OF AVERAGING MODELS

**Definition 18** (Uniform Stability, Definition 2.1 in Hardt et al. (2016)). *Let $f(h, z)$ denote a loss function on hypothesis $h$ and instance $z$. A randomized algorithm A is $\epsilon$-uniformly stable if for all datasets $S, S' \in \mathcal{Z}^n$ of size $n$ such that $S$ and $S'$ differ in at most one example, we have*

$$\sup_z \mathbb{E}_A \left[ f(A(S); z) - f(A(S'); z)) \right] \leq \epsilon_{stab}$$

**Theorem 19** (Averaging models is uniformly stable). *Given a set of users $U^{(i)} \in \mathcal{U}$ each with a local data set $D^{(i)} \in \mathcal{Z}$ originating from an unknown data distribution $\mathcal{Z}$, a learning algorithm $T$ with a $\Lambda$-strongly convex, $L$-Lipschitz, and $\beta$-smooth training objective $\mathcal{J}(f, D^{(i)}, K)$ on model parameters $f$ (like SGD_SVM of Alg. 1), an averaging routine $avg(T(\mho)) = \frac{1}{|\mathcal{U}|} \sum_{i=1}^{|\mathcal{U}|} T(D^{(i)}, \xi, K)$ with $\mho := \bigcup_i^{|\mathcal{U}|} D^{(i)}$ (like in Alg. 2), and the projected SGD update routine for a $c$-norm clipped data point $z_m^{(i)} \in D^{(i)}$ and class $k \in K$, i.e. $f_{m+1}^{(i)} = \prod_{\|f\| \leq R} \left( f_m^{(i)} - \alpha_t \frac{\partial}{\partial f} \mathcal{J}(f_m^{(i)}, z_m^{(i)}, k) \right) =: G$, then for a constant learning rate $\alpha \leq 1/\beta$, $M$ steps, and $N := |\mho|$ total data points, $T$ is $\epsilon_{stab}$-uniformly stable in the sense of Bousquet & Elisseeff (2002) with*

$$|\mathbb{E}_{D,SGD\_SVM}[\mathcal{J}(avg(T(\mho)), \mho, \_) - \mathbb{E}_{z \in \mathcal{Z}}[\mathcal{J}(avg(T(\mho)), z, \_)]]| \leq \epsilon_{stab} \leq \frac{2L^2}{\Lambda N} \in \mathcal{O}(N^{-1}).$$

*Proof.* By definition of uniform stability (Hardt et al., 2016, Definition 2.1), it suffices to prove (cf. Hardt et al. (2016, Theorem 2.2))

$$\sup_{z,k} \mathbb{E}_T \left[ \mathcal{J}(\frac{1}{|\mathcal{U}|} \sum_{i=1}^{|\mathcal{U}|} f_M^{(i)}, z, k) - \mathcal{J}(\frac{1}{|\mathcal{U}|} \sum_{i=1}^{|\mathcal{U}|} f_M^{'(i)}, z, k) \right] \leq \epsilon_{stab}$$

with $f_M^{(i)} = T(D^{(i)}, \xi, k)$ and $f_M^{'(i)} = T(D'^{(i)}, \xi, k)$ respectively after $M$ steps where $\bigcup_i D^{(i)}, \bigcup_i D'^{(i)}$ are 1-neighboring datasets.

We know due to the Lipschitz condition that for a given $z, k$

$$\mathbb{E} \left[ \mathcal{J}(\frac{1}{|\mathcal{U}|} \sum_{i=1}^{|\mathcal{U}|} f_M^{(i)}, z, k) - \mathcal{J}(\frac{1}{|\mathcal{U}|} \sum_{i=1}^{|\mathcal{U}|} f_M^{'(i)}, z, k) \right] \leq L \, \mathbb{E}[\delta_M].$$

with $\delta_m = \left\| \frac{1}{|\mathcal{U}|} \sum_{i=1}^{|\mathcal{U}|} f_m^{'(i)} - f_m^{(i)} \right\| \leq \frac{1}{|\mathcal{U}|} \sum_{i=1}^{|\mathcal{U}|} \left\| f_m^{'(i)} - f_m^{(i)} \right\|$.

Next, we need to bound $\mathbb{E}[\delta_M]$ by defining a modified growth recursion (Hardt et al., 2016, Lemma 2.5) for two arbitrary sequences of gradient updates $G_1, \ldots, G_M$ and $G_1', \ldots, G_M'$, the starting point $f_0^{(i)} = f_0^{'(i)}$, any $i \in [1, |\mathcal{U}|]$, and some $j \in [1, |\mathcal{U}|]$ as

$$\delta_0 = 0$$

$$\delta_{m+1} \leq \begin{cases} \eta \delta_m & \text{if } G_m^{(i)} = G_m^{'(i)} \text{ is } \eta\text{-expansive} \\ \eta \delta_m + \frac{2\sigma_m}{|\mathcal{U}|} & \text{if } G_m^{(j)} \text{ and } G_m^{'(j)} \text{ are } \sigma\text{-bounded}, G_m^{(i)} \text{ is } \eta\text{-expansive} \end{cases}.$$

Note that we consider the differing element occurring only in one local gradient update and not in each one. We recall the definition of a gradient update as $f_{m+1} = G_m(f_m)$ and $f_{m+1}' = G_m'(f_m')$.

*Proof, growth recursion (case I).*

$$\begin{aligned}
\delta_{m+1} &= \frac{1}{|\mathcal{U}|} \sum_{i=1}^{|\mathcal{U}|} \left\| G_m(f_m'^{(i)}) - G_m(f_m^{(i)}) \right\| \\
&\leq \frac{1}{|\mathcal{U}|} \sum_{i=1}^{|\mathcal{U}|} \eta \left\| f_m'^{(i)} - f_m^{(i)} \right\| \qquad \text{(Expansiveness, cf. Hardt et al. (2016, Definition 2.3))} \\
&\leq \eta \delta_m.
\end{aligned}$$

*Proof, growth recursion (case II).*

$$\begin{aligned}
\delta_{m+1} &= \frac{1}{|\mathcal{U}|} \sum_{i=1,j\neq i}^{|\mathcal{U}|} \left\| G_m(f_m^{(i)}) - G_m(f_m'^{(i)}) \right\| + \frac{1}{|\mathcal{U}|} \left\| G_m(f_m^{(j)}) - G_m'(f_m'^{(j)}) \right\| \\
&= \frac{1}{|\mathcal{U}|} \sum_{i=1,j\neq i}^{|\mathcal{U}|} \left\| G_m(f_m^{(i)}) - G_m(f_m'^{(i)}) \right\| \\
&\quad + \frac{1}{|\mathcal{U}|} \left\| G_m(f_m^{(j)}) - G_m(f_m'^{(j)}) + G_m(f_m'^{(j)}) - G_m'(f_m'^{(j)}) \right\| \\
&\leq \frac{1}{|\mathcal{U}|} \sum_{i=1,j\neq i}^{|\mathcal{U}|} \left\| G_m(f_m^{(i)}) - G_m(f_m'^{(i)}) \right\| \\
&\quad + \frac{1}{|\mathcal{U}|} \left\| G_m(f_m^{(j)}) - G_m(f_m'^{(j)}) \right\| + \frac{1}{|\mathcal{U}|} \left\| G_m(f_m'^{(j)}) - G_m'(f_m'^{(j)}) \right\| \\
&\leq \frac{1}{|\mathcal{U}|} \sum_{i=1}^{|\mathcal{U}|} \left\| G_m(f_m^{(i)}) - G_m(f_m'^{(i)}) \right\| \\
&\quad + \frac{1}{|\mathcal{U}|} \left\| f_m'^{(j)} - G_m(f_m'^{(j)}) \right\| + \frac{1}{|\mathcal{U}|} \left\| f_m'^{(j)} - G_m'(f_m'^{(j)}) \right\| \\
&\leq \eta \delta_m + \frac{2\sigma_m}{|\mathcal{U}|} \qquad \text{(Expansiveness and } \sigma\text{-boundedness, cf. Hardt et al. (2016, Definition 2.3)).}
\end{aligned}$$

Having established the growth recursion, we now combine the bounds with their probability of occurrence as well as calculate the corresponding $\eta$-expansiveness and $\sigma$-boundedness terms. Hardt et al. (2016, Lemma 3.3) have shown that we have $\alpha L$-boundedness for a $L$-Lipschitz objective function as well as $(1-\alpha\Lambda)$-expansiveness for a learning rate $\alpha \leq 1/\beta$ and a $\beta$-smooth and $\Lambda$-strongly convex objective function (cf. Hardt et al. (2015, Proof of Theorem 3.9)). Since each user samples during each training iteration one data point, we have for a given iteration a probability of $|\mathcal{U}|/N$ that an individual data point of $\mho$ has been chosen resulting in differing gradient updates $G_m^{(i)} \neq G_m'^{(i)}$. Thus, we have

$$\begin{aligned}
\mathbb{E}[\delta_{m+1}] &\leq (1 - \frac{|\mathcal{U}|}{N})\eta \, \mathbb{E}[\delta_m] + \frac{|\mathcal{U}|}{N}(\eta \, \mathbb{E}[\delta_m] + \frac{2\sigma_m}{|\mathcal{U}|}) \\
&\leq (1 - \frac{|\mathcal{U}|}{N})(1 - \alpha\Lambda) \, \mathbb{E}[\delta_m] + \frac{|\mathcal{U}|}{N}(1 - \alpha\Lambda) \, \mathbb{E}[\delta_m] + \frac{|\mathcal{U}|}{N}\frac{2\alpha L}{|\mathcal{U}|} \\
&= (1 - \alpha\Lambda) \, \mathbb{E}[\delta_m] + \frac{2\alpha L}{N}.
\end{aligned}$$

The remaining part goes by the proof of Hardt et al. (2015, Theorem 3.9) with the Lipschitzness of the training objective as well as the growth recursion $\mathbb{E}[\delta_{m+1}]$.

In short, we unfold the recursion:

$$\mathbb{E}[\delta_M] \leq \frac{2L\alpha}{N} \sum_{m=1}^{M} (1 - \alpha\Lambda)^m \leq \frac{2L}{\Lambda N}$$

and insert it into our initial bound which gets us for all $k$ and any $z$

$$\mathbb{E}\left[\mathcal{J}(\frac{1}{|\mathcal{U}|}\sum_{i=1}^{|\mathcal{U}|}f_M^{(i)}, z, k) - \mathcal{J}(\frac{1}{|\mathcal{U}|}\sum_{i=1}^{|\mathcal{U}|}f_M'^{(i)}, z, k)\right] \le \frac{2L^2}{\Lambda N}.$$

$\square$

Note that this proof permits the learning rate scheduling in Secure Distributed DP-Helmet which has been set to $\alpha_m := \min(1/\beta, 1/\Lambda m)$ for iteration $m$ and the $\beta$-smooth and $\Lambda$-strongly convex objective.

# N CONVERGENCE OF AVERAGING MODELS

**Definition 20** (Convergence). *Let $f(h, z)$ denote a loss function on hypothesis $h$ and instance $z$ and $F_S(h) := \frac{1}{|S|}\sum_{z\in S} f(h, z)$ the empirical risk on some dataset $S$. An algorithm $A$ converges with rate $\epsilon_{conv}$ under a data distribution $Z$ if*

$$\mathbb{E}_{S\in Z}[F_S(A(S)) - \inf_h F_S(h)] \le \epsilon_{conv}.$$

**Theorem 21** (Averaging models converges). *Given a set of users $U^{(i)} \in \mathcal{U}$ each with a local data set $D^{(i)}$, a learning algorithm $T$ with a $\Lambda$-strongly convex, $L$-Lipschitz, and $\beta$-smooth training objective $\mathcal{J}(f, D^{(i)}, K)$ on model parameters $f$ (like SGD_SVM of Alg. 1), an averaging routine $avg(T(\mho)) = \frac{1}{|\mathcal{U}|}\sum_{i=1}^{|\mathcal{U}|} T(D^{(i)}, \xi, K)$ with $\mho := \bigcup_i^{|\mathcal{U}|} D^{(i)}$ (like in Alg. 2), and the projected SGD update routine for a $c$-norm clipped data point $z_m^{(i)} \in D^{(i)}$ and class $k \in K$, i.e. $f_{m+1}^{(i)} = \prod_{\|f\|\le R}\left(f_m^{(i)} - \alpha_t \frac{\partial}{\partial f}\mathcal{J}(f_m^{(i)}, z_m^{(i)}, k)\right) =: G$, then for a diminishing learning rate $\alpha_m = \min(\frac{1}{\beta}, \frac{1}{\Lambda m})$, $M$ steps, a given $Z := \left\|\frac{1}{|\mathcal{U}|}(\sum_{i=0}^{|\mathcal{U}|} f_0^{(i)}) - f_*\right\|$, and a bias term $b$, $T$ converges to $f_* := \mathrm{argmin}_f \mathcal{J}(f, \mho, \_)$ with*

$$\mathbb{E}[\mathcal{J}(avg(T(\mho)), \mho, \_) - \mathcal{J}(f_*, \mho, \_)] \le \epsilon_{conv} \le \frac{\beta L^2}{2\Lambda^2}(M-1)^{-1} + b(M-1)^{-2} \in \mathcal{O}(M^{-1}).$$

*The bias term $b$ depends on $Z, \beta, \Lambda, L$.*

*Proof.* The proof is based on Nemirovski et al.'s proof of convergence for strongly convex SGD_SVM training. (Nemirovski et al., 2009, Section 2.1). Subsequently, we abbreviate the output of the learning algorithm $T$ at iteration $m$ for the $i$-th user with $f_m^{(i)} := T(D^{(i)}, \xi, k)$. First, we define the convergence criterion $A_m$ at the iterate $m$ and then its recursive growth $A_{m+1}$.

$$A_m = \frac{1}{2}\mathbb{E}\left[\left\|\frac{1}{|\mathcal{U}|}(\sum_{i=0}^{|\mathcal{U}|} f_m^{(i)}) - f_*\right\|^2\right]$$

Our convergence criterion describes that we measure and subsequently seek to bound the difference in the weights between the averaged $T$'s $\frac{1}{|\mathcal{U}|}\sum_{i=0}^{|\mathcal{U}|} f_m^{(i)}$ and the optimal weights $f_*$ for the loss $\mathcal{J}$ on

the combined data of all users. Subsequently, we abbreviate $G(f) := \frac{\partial}{\partial f} \mathcal{J}(f, \_, \_)$.

$$A_{m+1} = \frac{1}{2}\,\mathbb{E}\left[\left\|\frac{1}{|\mathcal{U}|}(\sum_{i=0}^{|\mathcal{U}|}\Pi_{\|f\|\leq R}(f_m^{(i)} - \alpha_m G(f_m^{(i)}))) - f_*\right\|^2\right]$$

$$= \frac{1}{2}\,\mathbb{E}\left[\left\|\frac{1}{|\mathcal{U}|}(\sum_{i=0}^{|\mathcal{U}|}\Pi_{\|f\|\leq R}(f_m^{(i)} - \alpha_m G(f_m^{(i)}))) - \Pi_{\|f\|\leq R}(f_*)\right\|^2\right]$$

$$\leq \frac{1}{2}\,\mathbb{E}\left[\left\|\frac{1}{|\mathcal{U}|}(\sum_{i=0}^{|\mathcal{U}|}f_m^{(i)} - \alpha_m G(f_m^{(i)})) - f_*\right\|^2\right]$$

(binomial expansion $\langle x + y, x + y\rangle = \langle x, x\rangle + 2\langle x, y\rangle + \langle y, y\rangle$ and linearity of expectation)

$$= A_m + \frac{1}{2}\alpha_m^2\,\mathbb{E}\left[\left\|\frac{1}{|\mathcal{U}|}\sum_{i=0}^{|\mathcal{U}|}G(f_m^{(i)})\right\|^2\right] - \alpha_m\,\mathbb{E}\left[(\frac{1}{|\mathcal{U}|}(\sum_{i=0}^{|\mathcal{U}|}f_m^{(i)}) - f_*)^T(\frac{1}{|\mathcal{U}|}\sum_{i=0}^{|\mathcal{U}|}G(f_m^{(i)}))\right]$$

Recall that because of the $L$-Lipschitz continuity of $\mathcal{J}$, $\|G(f)\| \leq L$. Hence, $\mathbb{E}\left[\left\|\frac{1}{|\mathcal{U}|}\sum_{i=0}^{|\mathcal{U}|}G(f_m^{(i)})\right\|^2\right] = \mathbb{E}\left[\left\|G(\frac{1}{|\mathcal{U}|}\sum_{i=0}^{|\mathcal{U}|}f_m^{(i)})\right\|^2\right] \leq L^2$. We now have for the recursion

$$A_{m+1} \leq A_m - \alpha_m\,\mathbb{E}\left[(\frac{1}{|\mathcal{U}|}(\sum_{i=0}^{|\mathcal{U}|}f_m^{(i)}) - f_*)^T G(\frac{1}{|\mathcal{U}|}\sum_{i=0}^{|\mathcal{U}|}f_m^{(i)})\right] + \frac{1}{2}\alpha_m^2 L^2.$$

Recall, strong convexity states that $(f' - f)^T(\nabla\mathcal{J}(f') - \nabla\mathcal{J}(f)) \geq \mu\|f' - f\|^2, \forall f', f$. Hence, we also know for the optimal $f_*$ that $(f' - f_*)^T\nabla\mathcal{J}(f') \geq \mu\|f' - f_*\|^2, \forall f'$. With $f' := \frac{1}{|\mathcal{U}|}\sum_{i=0}^{|\mathcal{U}|}f_m^{(i)}$, we conclude

$$\mathbb{E}\left[(\frac{1}{|\mathcal{U}|}(\sum_{i=0}^{|\mathcal{U}|}f_m^{(i)}) - f_*)^T G(\frac{1}{|\mathcal{U}|}\sum_{i=0}^{|\mathcal{U}|}f_m^{(i)})\right] \geq \mu\,\mathbb{E}\left[\left\|\frac{1}{|\mathcal{U}|}(\sum_{i=0}^{|\mathcal{U}|}f_m^{(i)}) - f_*\right\|^2\right] = 2\mu A_m.$$

The strong convexity constant $\mu = \Lambda$ can be determined by $\mathbf{H}\left(J(f, \bigcup_i D^{(i)}, \_)\right) \succeq \Lambda I, \forall f$ where $\mathbf{H}$ is the hessian matrix, $I$ the identity matrix and $B \succeq \iota I$ means that $B - \iota I$ is positive semi-definite. As the argumentation above about $J$'s strong convexity holds for any $f$, it also holds for $f = \frac{1}{|\mathcal{U}|}\sum_{i=0}^{|\mathcal{U}|}f_m^{(i)}$.

In summary, we now have

$$A_{m+1} \leq (1 - 2\Lambda\alpha_m)A_m + \frac{1}{2}\alpha_m^2 L^2. \tag{21}$$

The smoothness assumption can be equivalently formulated as $\|\nabla\mathcal{J}(f) - \nabla\mathcal{J}(f')\| \leq \beta\|f - f'\|, \forall f, f' \Leftrightarrow \mathcal{J}(f) \leq \mathcal{J}(f_*) + \frac{1}{2}\beta\|f - f_*\|^2, \forall f \Leftrightarrow \|\mathbf{H}(\mathcal{J}(f))\| \leq \beta, \forall f$. Similarly to the argumentation above, since beta smoothness holds for any $f$, it also holds for $f = \frac{1}{|\mathcal{U}|}\sum_{i=0}^{|\mathcal{U}|}f_m^{(i)}$. Thus we conclude that

$$\mathbb{E}\left[\mathcal{J}(\frac{1}{|\mathcal{U}|}\sum_{i=0}^{|\mathcal{U}|}f_M^{(i)}, \mho, \_) - \mathcal{J}(f_*, \mho, \_)\right] \leq \frac{1}{2}\beta\,\mathbb{E}\left[\left\|\frac{1}{|\mathcal{U}|}(\sum_{i=0}^{|\mathcal{U}|}f_M^{(i)}) - f_*\right\|^2\right] = \beta A_M.$$

By unraveling the recursive formula of $A_M$ (cf. Equation (21)) we get with the base case $A_0$

$$\leq \beta\left(\sum_{m=1}^{M}\left(\prod_{n=m+1}^{M}1 - 2\Lambda\alpha_n\right)\frac{1}{2}\alpha_m^2 L^2\right) + \beta\left(\prod_{n=1}^{M}1 - 2\Lambda\alpha_n\right)A_0.$$

Recall the learning rate $\alpha_m = \min(\frac{1}{\beta}, \frac{1}{\Lambda m})$ At $m_0 = \frac{\beta}{\Lambda}$ are we switching the learning rate from $\frac{1}{\beta}$ to $\frac{1}{\Lambda m}$. First, we consider the case $m \leq m_0$ where we rewrite for a constant learning rate $\alpha_m = \frac{1}{\beta}$ and $\varphi = \frac{L^2}{2\beta}$:

$$
\mathbb{E}\left[\mathcal{J}(\frac{1}{|\mathcal{U}|}\sum_{i=0}^{|\mathcal{U}|} f_{m_0}^{(i)}, \mho, \_) - \mathcal{J}(f_*, \mho, \_)\right] \leq \varphi\left(\sum_{m=1}^{m_0}\left(\prod_{n=m+1}^{m_0} 1 - \frac{\Lambda}{\beta}\right)\right) + \beta\left(\prod_{n=1}^{m_0} 1 - \frac{\Lambda}{\beta}\right) A_0
$$

$$
\leq \varphi\left(\sum_{m=1}^{m_0}(1 - \frac{\Lambda}{\beta})^{m_0-m}\right) + m_0(\beta - \Lambda)A_0 =: b'
$$

Next, we consider the case $m > m_0$ where we rewrite for a diminishing learning rate $\alpha_m = \frac{1}{\Lambda m}$ as well as $\varsigma = \frac{\beta L^2}{2\Lambda^2}$

$$
\mathbb{E}\left[\mathcal{J}(\frac{1}{|\mathcal{U}|}\sum_{i=0}^{|\mathcal{U}|} f_M^{(i)}, \mho, \_) - \mathcal{J}(f_*, \mho, \_)\right]
$$

$$
\leq \varsigma\left(\sum_{m=m_0+1}^{M}\left(\prod_{n=m+1}^{M} \frac{n-2}{n}\right)\frac{1}{m^2}\right) + \left(\prod_{l=m_0+1}^{M} \frac{l-2}{l}\right) b'
$$

$$
\leq \varsigma\left(\sum_{m=m_0+1}^{M}\left(\prod_{n=m+1}^{M} \frac{n-2}{n}\right)\frac{1}{m^2}\right) + \frac{(m_0-1)m_0}{(M-1)M}b'
$$

$$
= \varsigma\frac{\sum_{m=m_0+1}^{M} 1 - \frac{1}{m}}{(M-1)M} + \frac{1}{(M-1)M}(m_0-1)m_0 b'
$$

$$
\leq \varsigma\frac{M}{(M-1)M} + \frac{1}{(M-1)M}\underbrace{((m_0-1)m_0 b' - m_0)}_{=:b}
$$

$$
\leq \varsigma(M-1)^{-1} + b(M-1)^{-2}
$$

If we approach $M$ to $\infty$, i.e. assume a sufficient number of iterations, we further simplify

$$
\lim_{M\to\infty} \mathbb{E}\left[\mathcal{J}(\frac{1}{|\mathcal{U}|}\sum_{i=0}^{|\mathcal{U}|} f_M^{(i)}, \mho, \_) - \mathcal{J}(f_*, \mho, \_)\right] = 0
$$

which proves the convergence.

$\square$

Note that the bias term $b$ depends on how many iterations are conducted with the constant learning rate $1/\beta$.

