# OpenReview forum: "Single SMPC Invocation DPHelmet: Differentially Private Distributed Learning on a Large Scale"
_ICLR.cc/2023/Conference — Submitted to ICLR 2023_

### Official Review · Reviewer_a5Qe · 2022-10-15

**Confidence:** 3
**Correctness:** 2
**Technical Novelty And Significance:** 2
**Empirical Novelty And Significance:** 2
**Recommendation:** 3

**Clarity, Quality, Novelty And Reproducibility:**

The clarity could be improved. Here are two suggestions:

1. Use a table or figure to directly compare with alternative options and highlight the position of this paper.

2. Try to make the Overview section more concise. There are 10 unorganized paragraphs in the Overview, each highlighting a different point. It’s hard for me to get a clear overview.

**Strength And Weaknesses:**

To better understand the position of this paper, here are two alternative options in the literature.

**Option A**. Each user perturbs the local gradient and then sends the noisy gradient to the server. The serve then broadcasts the aggregated gradient. Such communication is repeated for many times. This approach has a high communication cost because there are many rounds. The variance of noise is also large because it depends on the local data size $n_i$ instead of overall data size $N$ (bad utility, bad communication cost, the baseline in this paper).

**Option B**. All users use SMPC to send noisy gradients to server, the noise variance in aggregated gradient is small because it scales with $1/N$ instead of $1/n_{i}$. The noise variance depends on overall data size, but the communication cost is even higher because 1) one still need to run multiple communication rounds; 2) each call of SMPC is more expensive than standard communication and the cost of SMPC scales with the number of users (good utility, bad communication cost, not evaluated in this paper).

The authors try to design an algorithm that is good in terms of both utility and communication cost. The main design choices are 1) they use an advanced SMPC protocol (Bell et al. 2020), which could scale to millions of users while the protocol used by previous works cannot; 2) they only call the SMPC once; 3) in order to maintain reasonable accuracy, they use a pre-trained feature extractor and only train local linear classifiers. The authors show the proposed algorithm is significantly faster and, in some cases, achieves higher accuracy than Option A.

In general, I think this work is a good attempt towards solving an important problem, i.e., making the algorithm good in both criterions. However, there is still room for improvement. Please see my comments/suggestions below.

1. An important baseline is missing. This paper does not evaluate the algorithms take Option B, e.g., [1]. Note that those algorithms could also scale to millions of users with the advanced protocol. A fair comparison would be all algorithms use the same protocol. In terms of efficiency, such a comparison would demonstrate the efficiency advantage of calling only a single SMPC. In terms of utility, it is important to show the drop in accuracy. The drop in accuracy seems non-negligible. With a pretrained feature extractor+linear classifier and eps=0.5, [2,3] both achieve ~90% accuracy on CIFAR-10 (see Figure 5 in [2] and Table 3 in [3]) while the accuracy in this work is $<$70%.

2. Although the noise scales with $1/N$, the utility guarantee of the aggregated model would not scale with $1/N$ because each local minimizer only sees $n_i$ datapoints. This may be inevitable if you only allow a single SMPC. However, if you run SMPC $log(N)$ times, it is possible to have a global utility guarantee scales with $1/N$. A simple example is Option 2 with DP-GD, which is studied in [1]. I think a good distributed private learning algorithm should have a utility bound scales with $1/N$. If the authors are interested in this direction, I would recommend papers about variance reduced methods in the non-private distributed learning setting, e.g., [4]. The basic idea is to let each user minimize its local objective (inner loop) and only occasionally run synchronization (outer loop), which would result in fewer calls of SMPC than DP-GD.

3. The utility of the local minimizer may be improved with gradient perturbation (DP-SGD or DP-GD). The current version uses output perturbation on the minimizer of a regularized Huber loss. Note that DP-SGD and DP-GD usually achieve better empirical performance (see [5]) despite they add noise at every step. The reason may be the noises added in earlier iterations get corrected by later gradient updates.

[1]: Distributed Learning without Distress: Privacy-Preserving Empirical Risk Minimization, https://proceedings.neurips.cc/paper/2018/hash/7221e5c8ec6b08ef6d3f9ff3ce6eb1d1-Abstract.html.

[2]: Differentially Private Learning Needs Better Features (or Much More Data),
https://arxiv.org/abs/2011.11660.

[3]: Do Not Let Privacy Overbill Utility: Gradient Embedding Perturbation for Private Learning, https://arxiv.org/abs/2102.12677.

[4]: Convergence of Distributed Stochastic Variance Reduced Methods without Sampling Extra Data, https://arxiv.org/abs/1905.12648.

[5]: A Practitioners Guide to Differentially Private Convex Optimization, https://icml.cc/Conferences/2021/ScheduleMultitrack?event=12667. You can find the presentation at https://slideslive.com/38964861/contributed-talks-session-2?ref=folder-87822 (starts ~32:20).


**Summary Of The Paper:**

This paper works on the efficiency of distributed differentially private learning without a trusted server. The main idea is that each user first trains the local model. Then the sever aggregates the local minimizers with only a single call of Secure Multi-Party Computation (SMPC).

**Summary Of The Review:**

My recommendation is mainly based on 1) one important baseline is missing; 2) there is neither strong empirical performance nor a reasonable theoretical utility bound. Please see Strength And Weaknesses for possible improvements.

---

> ### Author Response · Authors · 2022-11-18
> **Rebuttal Response**
>
> We thank the reviewers for the insightful comments. Below, we will respond to some specific comments and refer to the shared comment above for points that overlap with other reviews.
>
> ## Utility guarantee
> We do prove a utility bound of $\mathcal{O}(1/nm)$ to the population optimum with $nm$ being the combined number of data points across all users, which is the same as for central learning. In detail, we have shown uniform stability and convergence where we are able to show that averaged SVMs converge to a centrally trained SVM which sees all data points (cf. Section 4, “Stability & Convergence”). We do not need any additional synchronization steps for that. Uniform stability shows us that we are generalizing the same as a centrally trained model, or put differently: we generalize linearly to the number of involved users.
>
> ## Imprecise Baselines
> We mainly compare ourselves to “Option A” (Each user perturbs the local gradient with independent noise) since it is in our view the closest alternative approach. While “Option B” (All users use SMPC to send noisy gradients to the server) is interesting as well, we sought after a use case with very low computational and communication overhead. Thus we discussed these options which can boost privacy or utility by introducing more communication rounds or more SMPC invocations only in the introduction/related work but did not include these in our experiments.
>
> We clarified the possible baselines in our paper and included a new tabular comparison (cf. Table 1).
>
> ## Drop in accuracy
> We have run our experiments more excessively and now report more favorable numbers. Previously, we had for $1{,}000$ users and $\varepsilon=0.5$ a classification accuracy of ~$64\\,\\%$, now we can report ~$77\\,\\%$. Additionally, to provide a fair comparison, we accounted in the distributed setting for dishonest users which led to an increased noise scale of $2$ in the averaged model. When accounting for this fact as well as for our convergence bounds which showed that we converge to a centrally trained model, we can read the numbers for $\varepsilon=0.5$ and $\mathit{n\\_users}=1$ instead: ~$87\\,\\%$ (for $1$ user we do not have the increased noise scale of $2$ since in this setting, dishonest users do not make much sense).
>
> We suspect to account for the remaining accuracy drop to the involved SVM since [Reb3, Reb4] leverages differentially private GD-based schemes. We observe on CIFAR-10 and for $\varepsilon = 0.59$, that DP_SGD_SVM has an accuracy of $87.4\\,\\%$ while DP-SGD has $93.6\\,\\%$:
> The reasons include leakage via sequential composition (through DP-SGD-SVM’s one-versus-rest multi-class approach) compared to DP-SGD’s joint learning of all classes as well as its noise-correcting property from its iterative noise application.
>
> We have explained the drop in accuracy better in an updated version of our paper.
>
> [Reb3]: Florian Tramèr and Dan Boneh. “Differentially private learning needs better features (or much more data)”. In International Conference on Learning Representations (ICLR), 2021.
>
> [Reb4]: Yu, Da, Huishuai Zhang, Wei Chen, and Tie-Yan Liu. "Do not let privacy overbill utility: Gradient embedding perturbation for private learning." In International Conference on Learning Representations (ICLR), 2021.
>
> ## Variance-reduced methods by global synchronization steps
> The work by [Reb5] assumes that the aggregator randomly shuffles the data points among a set of users. This contradicts our attacker model and security properties: we assume that each party does not want to reveal its local data to the other parties. Thus the results of [Reb5] cannot be applied to our setting.
> A privacy-preserving mechanism along the lines of [Reb5] is an interesting direction for future work.
>
> [Reb5] Cen, Shicong, Huishuai Zhang, Yuejie Chi, Wei Chen, and Tie-Yan Liu. "Convergence of distributed stochastic variance reduced methods without sampling extra data." IEEE Transactions on Signal Processing 68 (2020): 3976-3989.

---

> > ### Comment · Reviewer_a5Qe · 2022-11-21
> > **Continued discussion**
> >
> >
> > Thank you for your response. I appreciate the efforts. Please find my reply to the first to points in your comment to all reviewers.
> >
> > **Drop in accuracy**
> >
> > I don't think the theoretical bound implies the accuracy of $n_user=1$ can be read as the accuracy of $n_user>1$. When there are no dishonest users, what is the trend of accuracy (in practice) if you keep increasing the number of users?
> >
> > **Variance-reduced methods by global synchronization steps**
> >
> > My intention was that you can use more calls of SMPC to improve the accuracy and the analysis in the reference may be helpful (although you may need stronger assumption than they do).

---

> > > ### Author Response · Authors · 2022-11-24
> > > **Continued Discussion**
> > >
> > > Thank you for your additional comments!
> > > ### Drop in accuracy
> > > Our convergence results show that DPHelmet converges to the optimum as in the centralized setting (cf the convergence criterion A_m in Theorem 21 at the beginning of the proof). Thus, we can report even in the distributed training scenario with $n>1$ user, the accuracy for $n=1$ user. This is also indicated by our experiments in Figure 3 (left) where the accuracy for $n=\\{100, 400, 1000\\}$ are close.
> > >
> > > Our convergence bound suggests that with even longer training (i.e. more epochs) this gap between $n=1$ and $n>1$ could be closed further. In particular, our convergence bounds specify the convergence rate for each user. Due to time restrictions in our experiments, the number of total SGD iterations is the same for each $n$. As a result, the larger the $n$, the fewer SGD iterations each user conducts. This is also supported experimentally, as a large difference between the DP-SGD-SVM experiments in our previous paper version from September and the current one was that we previously trained for $100$ instead of $500$ epochs. When comparing the plots (previously: Figure 2 (left); now: Figure 3 (left)), the gap closes.
> > >
> > > Note that for one user we are not close since we deactivated the dishonesty assumption which decreased the noise scale by a factor of $2$ (i.e., in the left part of Figure 3 you would have to approximately half the epsilon for the $n>1$ DP-SGD-SVM plots).

---

### Official Review · Reviewer_qk3x · 2022-10-24

**Confidence:** 4
**Correctness:** 2
**Technical Novelty And Significance:** 2
**Empirical Novelty And Significance:** 2
**Recommendation:** 3

**Clarity, Quality, Novelty And Reproducibility:**

the main idea of using secure aggregation to bypass utility challenges in federated learning with DP seems new (though there might be papers i haven't read which does something similar).

the paper can improve on its writing clarity. given the aforementioned technical and writing issues, it's hard to verify the experimental results.

**Strength And Weaknesses:**

strength: the idea of introducing secure aggregation into private machine learning is interesting. this idea likely helps bypass certain limitations of federated learning under local DP in terms of utility (albeit introducing new, but perhaps mild, assumptions, e.g., at least half the users are honest).

weakness: while the main idea makes sense, there are several technical as well as presentational issues which lead to the paper being hard to read. in addition, some of the results / claims are hard to verify (and i personally doubt some claims are true). i list the main points below.

- it's unclear what model of a "dishonest user" this work assumes. for instance, in distributed computing, there's this notion of byzantine nodes, which essentially means that they can do whatever they'd like. in milder models, a dishonest user might just be assumed to fail without tampering with aggregation (but also not provide their data). i'm assuming the claim that ">50% users being honest implies secure aggregation works" is dependent on a precise notion of a "dishonest user" whose definition which i was not able to pinpoint in the main text. in the extreme case, what if i assumed the dishonest user is omnipotent and was able to "steal" the key of all honest users? wouldn't that give the dishonest user the ability to revert the noise?

- algorithm 1 seems problematic in several ways.
  - the loss defined on 2 is independent of datapoints whose y is not k. this would mean that the models trained here are only increasing the margin for the positive class -- the margins for the negative class is not penalized, and can grow however they like. there's a chance i might be missing something, but i'd be very surprised if this loss gives high-performing models. can authors clarify how they train these models and perform inference?
    - personally, i'd perhaps use the loss J = L2 reg + average of l_huber ( f clipped(x) [ 1[y = k] - 1[y \ne k] ] ), which takes the negative classes' margin into account.
  - for privacy purposes, i can somewhat see why the loss on line 2 is desirable -- it ensures that each data point only affects the loss of a single model. one could probably argue that this gives a smaller sensitivity (than naive joint release, which produces a \sqrt{K} factor), though this nuance is not explicitly mentioned in the main text. more generally, if one uses the loss i defined above (the one that penalizes small margins for both pos and neg classes), one cannot assume the joint release still has sensitivity s, but would rather need the conservative bound of \sqrt{K} s (s is sensitivity of single model release for a particular k). section 3.1 in the main text does not make these points explicit.
  - have the authors tried the simpler logistic regression? the loss for multi-class logistic regression is strongly convex, and lipschitz (for bounded iterates), and does not possess this awkward one-vs-rest problem with SVMs.

- the authors frame their method with training linear models as an integral component. note while the framework is generally applicable to strongly convex and lipschitz losses (where coefficients are known), the overall method is limited to convex problems, and cannot be applied to when linear model adaptions cannot work well.

- given the strong resemblance to early works such as PATE, i'm surprised the similarities and differences are not discussed. i personally can figure out these, but this job should not be left to the reader solely. some references below.

Papernot, Nicolas, et al. "Scalable private learning with pate." arXiv preprint arXiv:1802.08908 (2018).
Papernot, Nicolas, et al. "Semi-supervised knowledge transfer for deep learning from private training data." arXiv preprint arXiv:1610.05755 (2016).
Choquette-Choo, Christopher A., et al. "Capc learning: Confidential and private collaborative learning." arXiv preprint arXiv:2102.05188 (2021).


**Summary Of The Paper:**

the paper proposes a method that combines transfer learning (based on fitting linear model on pretrained features), secure aggregation, and differentially private mechanisms, to achieve secure and private machine learning. authors empirically compare against DP-FL and claim performance improvements in the large user regime.

**Summary Of The Review:**

the paper proposes a method that combines transfer learning (based on fitting linear model on pretrained features), secure aggregation, and differentially private mechanisms, to achieve secure and private machine learning. while interesting, the works have several technical and writing issues that can be improved on.

---

> ### Author Response · Authors · 2022-11-18
> **Rebuttal Response**
>
> We thank the reviewers for the insightful comments. Below, we will respond to some specific comments and refer to the shared comment above for points that overlap with other reviews.
>
> ## Imprecise Algorithm 1
> Thank you for pointing out that bug. We have fixed it and it has not influenced the results of our experiments. We assume that the “negative” data points are considered for any given “positive” class. In detail, we meant that when the indicator function $1[y=k]$ is false, we have a negative “0”-labeled class.
>
> ## Privacy accounting for the number of classes
> Thank you for pointing out that bug. We have fixed it and it has not influenced the results of our experiments. In essence, we assume $K$ compositions of the Gaussian noise added to the SVM parameters which has an impact of roughly $\sqrt{K}$ on the noise scale. Via the PrivacyBucket Toolbox or Theorem 5 of Sommer et al. [Reb6] respectively, we are able to infer the correct and numerically tight ($\varepsilon$, $\delta$)-DP bounds.
>
> [Reb6] David M Sommer, Sebastian Meiser, and Esfandiar Mohammadi. Privacy loss classes: The central
> limit theorem in differential privacy. Proceedings on privacy enhancing technologies, 2019(2): 245–269, 2019.
>
> ## Dishonest users
> In a distributed training scheme like ours, there are two relevant vulnerabilities posed by malicious users: First, an untrustworthy noise generation and maliciously contributed data, and second, misbehavior in the secure summation protocol.
>
> We do account for up to $t$ malicious users by letting honest users increase their noise scale by a factor of $\sqrt{1/t}$. Our security guarantees also allow maliciously contributed SVM models.
>
> If the application scenario enables an honest-but-curious, global, polynomial-time bounded network-level attacker, which follows the protocol, we can even use a highly scalable efficient secure summation protocol Bell et al. (2020). Note that their variant for malicious users leaks information about partial sums.
>
> To clarify, we updated the threat model in our paper.
>
> ## Limitation to convex learning problems
> While SVMs work best for convex learning problems, we showed in our experiments that we can still classify complex learning tasks like CIFAR-10 with very good classification performance thanks to effective pre-training. We suspect that pre-training applies to a huge variety of other datasets and learning tasks as well.
>
> Additionally, we also provided in Corollary 10 a variation of our method which works for arbitrary learning algorithms as long as they have a bounded model space. This additionally allows guarantees as in local-DP and shows very promising performance in our experiments for enough users (cf. Figure 2, previously Figure 1).
>
> ## Similarities to PATE
> Due to space constraints we did not discuss PATE, as it considers a very different application scenario where a medium-sized set of private data points is available and a large set of data points from the same distribution shall be labeled (e.g., for training another model on them). PATE is not a distributed privacy-preserving learning method while Secure Distributed DP Helmet is.
>
> ## Verify experimental results
> We also have provided code as supplemental material in our original submission.

---

### Official Review · Reviewer_p4WT · 2022-10-24

**Confidence:** 3
**Correctness:** 4
**Technical Novelty And Significance:** 2
**Empirical Novelty And Significance:** 3
**Recommendation:** 3

**Clarity, Quality, Novelty And Reproducibility:**

The presentation of the paper needs to be improved.
The reproducibility of the paper is good.

**Strength And Weaknesses:**

Strengths:
1. The problem of achieving privacy protection in distributed learning is timely and important, especially over large number of users.
2. Considered the combination of pre-trained model and private linear models to improve the model performance and communication efficiency.
3. Provided the uniform stability and convergence of proposed methods.

Weaknesses
1. The presentation of the proposed framework is not clear. The paper is quite hard to understand what kind of framework it proposed. It is better to give some figures for the framework and how the SimCLR-based pretraining is integrated in the whole pipeline.
2. Many related works on SMPC + DP are ignores and do not provide comparisons in the experiments, e.g., [1] Jayaraman, Bargav, Lingxiao Wang, David Evans, and Quanquan Gu. "Distributed learning without distress: Privacy-preserving empirical risk minimization." Advances in Neural Information Processing Systems 31 (2018).
3. The reason for introducing the computational DP is not clear.
4. How the pretrained model affects the model performance is not clear.
5. Why is the dimension of Gaussian variance $(p+1)\times K$ instead $p \times K$?
6. Please provide the utility and privacy tradeoff for the private version of Algorithm 1 and the comparison with common private erm methods, e.g., Wang, Di, Minwei Ye, and Jinhui Xu. "Differentially private empirical risk minimization revisited: Faster and more general." Advances in Neural Information Processing Systems 30 (2017).

**Summary Of The Paper:**

This paper proposes a scalable differentially private distributed learning framework for protecting privacy over distributed dataset. The scalability of the framework is based on a scalable secure summation protocol.  The privacy of the framework is achieved by aggregating the noise of non colluded users in an oblivious way instead of distributed noise generation over millions of users. They also consider a combination of SimCLR pre-training with linear convex private learning (DP-SVM) to achieve better performance.


**Summary Of The Review:**

The paper proposed a scalable differentially private distributed learning framework. However, the idea of integrating pre-trained model extractor with private convex model is not new. Also, the combination of DP with SMPC is also not new.

---

> ### Author Response · Authors · 2022-11-18
> **Rebuttal Response**
>
> We thank the reviewers for the insightful comments. Below, we will respond to some specific comments and refer to the shared comment above for points that overlap with other reviews.
>
> ## How Pretraining boosts DP performance
> We refer to related work that discusses this point in more detail. [Reb3, Reb4, Reb7]
> [Reb7] in short: on CIFAR-10 and $\varepsilon=1$, a non-pretrained network reaches about $57\\,\\%$ and a pre-trained network about $95\\,\\%$.
>
> [Reb3] Florian Tramèr and Dan Boneh. “Differentially private learning needs better features (or much more data)”. In International Conference on Learning Representations (ICLR), 2021.
>
> [Reb4] Da Yu, Huishuai Zhang, Wei Chen, Tie-Yan Liu. “Do Not Let Privacy Overbill Utility: Gradient Embedding Perturbation for Private Learning”. In International Conference on Learning Representations (ICLR), 2021.
>
> [Reb7] Soham De, Leonard Berrada, Jamie Hayes, Samuel L Smith, and Borja Balle. “Unlocking High-Accuracy Differentially Private Image Classification through Scale”. In: arXiv:2204.13650 (2022).
>
> ## Computational DP
> We clarified the reasoning for Computational DP in Appendix A.1.
> We technically require computational DP because we have computationally bounded attackers due to the involved SMPC protocol. However, the resulting increase in $\delta$ is negligible.
>
>
> ## Dimension of Gaussian variance
> We not only noise the $p$-dimensional SVM coefficients but also the $1$-dimensional intercept (the bias).

---

### Official Review · Reviewer_7DLA · 2022-11-01

**Confidence:** 4
**Correctness:** 3
**Technical Novelty And Significance:** 2
**Empirical Novelty And Significance:** 1
**Recommendation:** 3

**Clarity, Quality, Novelty And Reproducibility:**

While the idea of reducing smpc invocations is interesting, the current paper doesn’t investigate this question in depth.
The key of the paper seems to be that SVM are used, but theory nor experiments seem to prove any novelty in this nor a relation to the number of SMPC invocations.

The text could be better organized to more clearly relate objective, method, experiment and conclusion.

**Details Of Ethics Concerns:**

—

**Strength And Weaknesses:**


The exact claimed contribution isn’t very clear.  Even though compared to the earlier NeurIPS submission the authors added a “contributions” paragraph in the introduction, (a) several items in the bullet list are not contributions themselves but more properties of a contribution described earlier, and (b) the description of the contribution doesnt specify clearly to what extent this contribution improves over the state of the art.

It is unclear to me why the presented algorithm could run with a single SMPC invocation.

As already stated in my neurips 2022 review, it is unfortunate that the whole paper focuses on a single dataset (Cifar-10).



**Summary Of The Paper:**

This paper studies how to reduce the number of MPC rounds in privacy preserving learning.

**Summary Of The Review:**

While the authors seem to know good techniques, i dont see how the current paper uses them for a coherent contribution.

---

> ### Author Response · Authors · 2022-11-18
> **Rebuttal Response**
>
> We thank the reviewers for the insightful comments. Below, we will respond to some specific comments and refer to the shared comment above for points that overlap with other reviews.
>
> ## Why do we only need one SMPC invocation?
> Our strong utility bounds (convergence in Theorem 21 and stability in Theorem 19) prove that a single SMPC invocation suffices. If we locally train a strongly convex ERM model via SGD (e.g., SVMs), weakly noise them, and then average them via secure summation, we get the same utility and privacy bounds as if the union of all local data sets would have been locally trained.
>
> In particular, we managed to improve upon the utility bounds of averaged strongly convex SGD-based SVMs compared to prior work [Reb1]. This scenario is called output perturbation by Jayaraman et al. [Reb1]. These improved bounds are possible for SGD-based strongly convex ERM training, e.g. support vector machines (SVMs) or logistic regression (LR). These models also have the property that we can commonly share the noise among the parties thus the overall noise scales gracefully (i.e. inverse linearly) with the number of users.
>
> [Reb1]’s gradient perturbation variant needs $\mathcal{O}(\mathit{n\\_iterations})$ many SMPC invocations for similar privacy and utility guarantees than our work and their output perturbation variant also only needs one SMPC invocation but leverages worse utility bounds (to the population optimum).
>
> Hence, the closest baseline which matches the utility and privacy guarantees of the most optimal solution, i.e. centralized training, is the gradient perturbation variant of [Reb1]. However, this method does need $\mathcal{O}(\mathit{n\\_iterations})$ SMPC invocations, while our method only needs $1$.
>
> We added Table 1 which explicitly compares our work to two relevant baselines: differentially private federated learning (DP-FL), and the work by Jayaraman et al. [Reb1] with its output perturbation as well as gradient perturbation variant.
>
> [Reb1] Jayaraman, Bargav, Lingxiao Wang, David Evans, and Quanquan Gu. "Distributed learning without distress: Privacy-preserving empirical risk minimization." Advances in Neural Information Processing Systems 31 (2018).

---

### Author Response · Authors · 2022-11-18
**Rebuttal Comment**

We thank the reviewers for their insightful comments.

We believe that the contribution of our work has not been presented as well as it could have been, leading to a few misconceptions. We address those points and clarify our contribution in this response and incorporated them into the paper accordingly.

As suggested by reviewer p4WT, we have added a figure showing the integration of the various techniques in a new schematic figure of our work (cf. Figure 1). As suggested by reviewer a5Qe, we added a tabular comparison with related work (cf. Table 1).


## Related Work by Jayaraman et al. (p4WT, a5Qe)
Thank you for pointing us to the work of Jayaraman et al. [Reb1]. This theoretical work indeed shows weaker versions of our utility and sensitivity results.

Qualitatively, Jarayaman et al. ignore potential privacy leakage from the optimization algorithms (such as SMO or SGD) whereas we do not: we analyze the leakage of SGD. More specifically, they only use a bound for the sensitivity of the optimum and not for the result of an optimization algorithm.

Quantitatively (cf. Table 1), for their output perturbation variant which matches the number of SMPC invocations to our work, they have a worse utility bound to the population optimum: they are in $\mathcal{O}(1/m)$ (where $m$ is the number of local data points) whereas we show that our work is in $\mathcal{O}(1/(nm))$ (where $n$ is the number of users), just as in the centralized training setting. Their gradient perturbation variant has similar privacy and utility bounds but needs considerably more SMPC invocations proportional to the number of training iterations.

[Reb1] Jayaraman, Bargav, Lingxiao Wang, David Evans, and Quanquan Gu. "Distributed learning without distress: Privacy-preserving empirical risk minimization." Advances in Neural Information Processing Systems 31 (2018).

## Clarified Contributions
As reviewers p4WT and 7DLA wondered about what precisely were our contributions we would like to take the space to clarify these since we suspect that these might be relevant to the others as well.

We use the combination of DP+SMPC+Pretraining to demonstrate a practical approach with two tangible contributions:

1. For SGD-based strongly convex ERM, we show utility bounds of $\\mathcal{O}(1/(nm))$, where nm is the combined number of data points across all users. Here, we average locally trained SVM models only once. Thus our distributed approach matches the utility and privacy bound of centralized training while only requiring little communication overhead. If we leverage the secure summation protocol of Bell et al. (2020), we only need one secure summation invocation with $4$ rounds and a communication cost of $\\mathcal{O}(\\log(\mathit{n\\_users}) + \mathit{model\\_size}$. Consequently, we do not need additional synchronization steps.
Confer Section 4, “Stability & Convergence” for more details.

2. We show how with enough data, guarantees as in local DP can be achieved, even without assumptions on the training algorithm beyond a norm-bounded parameter space: we protect the entire input of a user while achieving strong utility bounds (> $80\\,\\%$ test accuracy for CIFAR-10).


## Related Work on DP-ERM (p4WT, a5Qe)
Thank you for pointing us to additional related work on DP-ERM like Wang et al. [Reb2]. We reference that in an updated version of our paper.

We have an ablation study of different DP-ERM techniques in Appendix E. Since we require an output sensitivity for the distributed training scheme (such that the noise scales gracefully with the number of users) many DP-ERM solutions are naturally excluded.

In Appendix E, we also discuss objective perturbation which performs worse since the noise does not scale well with the 10 CIFAR-10 classes. The noise of our algorithm scales with roughly $\sqrt{10}$ whereas for objective perturbation a similar noise-saving is difficult since we do not have an output sensitivity.

Gradient perturbation, e.g. used by [Reb2], is not applicable for a simple reason. In DP Helmet, each user trains and noises a local model with weak privacy guarantees (by using less noise in the output perturbation). We show that after a secure summation step these weak privacy guarantees add up to strong privacy guarantees (cf. Lemma 7). We are not aware of any results that show a similar amplification for gradient perturbation.

[Reb2] Wang, Di, Minwei Ye, and Jinhui Xu. "Differentially private empirical risk minimization revisited: Faster and more general." Advances in Neural Information Processing Systems 30 (2017).

---

> ### Comment · Reviewer_a5Qe · 2022-11-21
> **Continued discussion from Reviewer a5Qe**
>
> Thank you for the detailed response. It’s nice to see you have added Figure 1 and Table 1 to improve the clarity. I see two of my concerns are also raised by other reviewers, so I put part of my response here.
>
> 1. Continued discussion on Jayaraman et al.
>
> Although the gradient perturbation variant in Jayarmman et al. needs the number of SMPC invocations proportional to the number of training iterations $T$, their Theorem 3.6 (DP-GD) shows $T=\mathcal{O}(log(nm))$ is sufficient. The authors need to demonstrate why the improvement from $log(nm)$ to $1$ is significant **in practice**. More specifically, for a reasonable target accuracy, how much faster is DP Helmet compared to DP-GD?
>
> 2. Continued discussion on the utility bound.
>
> Thank you for clarifying that you have utility bounds of $\mathcal{O}(1/(nm))$. I found this a bit counterintuitive because each local minimizer only sees the local data. Take the linear regression problem as an example, which has $(X^{T}X)^{-1}X^{T}y$ as the analytical solution ($X$ is the data matrix and $y$ is the label vector). I don’t think splitting the data into two parts and averaging the minimizers would give the same solution. Could you please explain more on the claim "For SGD-based strongly convex ERM, we show utility bounds of $\mathcal{O}(1/(nm))$"? What restriction does this bound make?
>
> 3. Summary
>
> I think this work is worth publishing if the authors can demonstrate either 1) empirically, there is a reasonable application scenario that DP Helmet is faster than DP-GD (for a reasonable target accuracy), 2) theoretically, there are some meaningful data distributions on which DP Helmet gives the same utility bound as DP-GD for (strongly) convex problems.
>
> If the authors still choose not to include DP-GD in their experiments. Then, in order to demonstrate the first point, **another approach is to find real world applications where DP-GD is not applicable** (in which case I believe the authors still need to redesign the experiments). For example, the users in real world applications may only be online for a short time period.  In such case it is inevitably that one can only make a single query to each user.

---

> > ### Author Response · Authors · 2022-11-24
> > **Continued Discussion (I)**
> >
> > Thanks for your quick and detailed response! We are happy to answer any further questions.
> >
> > ### Related Work by Jayaraman et al.
> > We agree that for Jayaraman et al.’s gradient perturbation variant less iterations (about $\mathcal{O}(\log(nm)$) suffice. That being said, we see great benefits of one communication round versus $\mathcal{O}(\log(nm))$ rounds:
> >
> > One communication round enables low-interaction systems where users can participate asynchronously and do not have to stay online until all parties submitted their SVMs. One example is to implement the secure aggregation with a classical secret sharing based SMPC protocol with two computation servers. Hence, low-interaction systems are more robust which has the effect that we can include more users which reduces the overall amount of DP noise and subsequently boosts the utility. These systems are especially useful for smartphone-based applications (i.e. users use Smartphone hardware) where simultaneous availability of all devices is hard to come by. Additionally, for one communication round, we also require fewer computational resources which are typically limited in a Smartphone-based application (e.g. to prevent unnecessary battery drainage or lagging).
> >
> > We will investigate the empirical performance of DP-GD in our scenario and keep you notified if we finish them in time.

---

> > ### Author Response · Authors · 2022-11-24
> > **Continued Discussion (II)**
> >
> > ### Utility bound
> > We believe that there are indeed many meaningful data distributions on which DP Helmet gives the same utility bound as DP-GD for (strongly) convex problems, namely: our guarantees hold for arbitrary data distributions. While discussing utility bounds based on data distributions is a valid approach, we follow another line of work that provides utility bounds independent of the data distributions and focuses on other criteria such as convergence and stability. Jayaraman et al. also provides utility bounds independent of the data distribution.
> >
> > > What restriction does this bound make?
> >
> > We have four requirements for our $\mathcal{O}(1/(nm))$ utility bound:
> > 1. strongly convex training objective,
> > 2. smooth training objective,
> > 3. Lipschitz continuous training objective, and
> > 4. SGD-based update routine.
> >
> > > Could you please explain more on the claim "For SGD-based strongly convex ERM, we show utility bounds of $\mathcal{O}(1/(nm))$"?
> >
> > In short: We provide a similar utility bound definition to the one used by Jayaraman et al. Our utility bound consists of two parts: uniform stability and convergence. For uniform stability, we reach the same bound as in the centralized setting (cf. Hardt et al., Theorem 3.9, [Reb8]). For convergence, we reach a similar bound per user as in the centralized setting (cf. e.g. Rakhlin et al., Theorem 1, [Reb9]) up to a constant factor and a bias of rate $\mathcal{O}(1/(\\#iterations)^2)$ induced by an additional learning rate restriction.
> >
> > Let us try to provide an intuition for our results: For uniform stability, we reach good bounds since an additional data point could only alter one of the $n$ SVMs. In each SVM, we have an empirical risk where we divide the per-data-point loss by $m$. Since we average the $n$ SVMs, the impact of any data point is thus $\mathcal{O}(1/(mn))$. For convergence, when unfolding the definition it turns out that the loss term for the averaged parameters (considered for the distributed SVMs) has the same constants for Lipschitzness, smoothness, and strong convexity as the loss term for the centralized setting. This means that the convergence behavior does not change compared to the centralized setting.
> >
> > --- long version ---
> >
> > We see this as an interesting discussion of what are useful and intuitive utility bounds. We defined the utility bounds in line with the literature (Hardt et al. [Reb8], Shalev-Shwartz et al. [Reb10]) as a bound consisting of a stability (“train-test generalization”) and a convergence part. Jayaraman et al. uses a similar definition.
> > We bound the difference of the population risk and the optimal empirical risk for an objective function $J$, an unknown data distribution $\mathcal{Z}$ where dataset $D \in \mathcal{Z}$, and the model parameters $f$:
> >
> > $E_{z \in \mathcal{Z}}[J(f, z)] - \inf_f J(f, D) \le \epsilon_{stab} + \epsilon_{convergence}$.
> >
> > Jayaraman et al. uses the bounds of Pathak et al. [Reb11] who have similar definition:
> >
> > $E_{z \in \mathcal{Z}}[J(f, z)] - E_{z \in Z} J(min_f J(f, D),z)$.
> >
> > Informally, both bounds differ as follows: the former bounds consider the distance of the population risk and the optimal *population* risk, and the latter bounds consider the distance of the population risk and the optimal *empirical* risk.
> >
> > Our bounds express that we reach the same empirical optimum as a centralized algorithm does (convergence, cf. Definition 20) and we can train-test generalize well by showing that the cost of forgetting a training data point for any users is small since the loss of the model would not change much (uniform stability, cf. Definition 18). Uniform stability thus shows that the empirical risk is close to the population risk. Since we also converge with $\mathcal{O}(1/\\#iterations)$ we can additionally show that the optimal empirical risk is close to the population risk as well. We also would like to stress that Shalev-Shwartz et al. [Reb10] connected convergence and a related stability notion with ERM learnability properties.
> >
> > [Reb8] Hardt, Moritz, Ben Recht, and Yoram Singer. "Train faster, generalize better: Stability of stochastic gradient descent." In International conference on machine learning, pp. 1225-1234. PMLR, 2016.
> >
> > [Reb9] Rakhlin, Alexander, Ohad Shamir, and Karthik Sridharan. "Making gradient descent optimal for strongly convex stochastic optimization." arXiv preprint arXiv:1109.5647 (2011).
> >
> > [Reb10] Shalev-Shwartz, Shai, Ohad Shamir, Nathan Srebro, and Karthik Sridharan. "Learnability, stability and uniform convergence." The Journal of Machine Learning Research 11 (2010): 2635-2670.
> >
> > [Reb11] Pathak, Manas, Shantanu Rane, and Bhiksha Raj. "Multiparty differential privacy via aggregation of locally trained classifiers." Advances in neural information processing systems 23 (2010).

---

### Decision · Program_Chairs · 2023-01-20

**Decision:**

Reject

**Justification For Why Not Higher Score:**

It's a long and messy paper which none of the reviewers liked

**Justification For Why Not Lower Score:**

N/A

**Metareview: Summary, Strengths And Weaknesses:**

This work proposes a learning algorithm in a federated setting that trains a (multi-class) linear SVM on local data of each client and then averages the resulting models via secure aggregation. A layer of differential privacy is achieved via distributed noise generation. To increase accuracy a pre-trained feature extractor is used (specifically SimCLR in this paper). This paper aims to support this approach via theory and empirical experiments. Unfortunately, as detailed in the reviews both results and their presentation appear to be flawed. At a high level
reducing distributed learning to simple averaging of model is an attractively simple idea. It's hardly new and, aside from some very special cases doesn't work well, in theory or practice. Most notably, in FL clients do not have samples from exactly the same distribution (and typically relatively few samples each). So averaging does not make theoretical or practical sense.